# Too much to handle? Interference from distractors with similar affordances on target selection for handled objects

**Lilas Haddad, Yannick Wamain, Solène Kalénine***

CNRS, UMR 9193 –SCALab–Sciences Cognitives et Sciences Affectives, Univ. Lille, Villeneuve-d'Ascq, France

* solene.kalenine@univ-lille.fr

## Abstract

The existence of handle affordances has been classically demonstrated using the Stimulus-Response Compatibility paradigm, with shorter response times when the orientation of the object handle and the response hand are compatible in comparison to incompatible. Yet the activation of handle affordances from visual objects has been investigated in very simple situations involving single stimulus and motor response. As natural perceptual scenes are usually composed of multiple objects that could activate multiple affordances, the consequence of multiple affordance activation on the perception and processing of a given object of the scene requires more investigation. The aim of this study was to determine the impact of distractor affordances on the processing of a target object in situations involving several familiar graspable objects. In two online experiments, 229 participants had to select a target object (the kitchen utensil or the tool) in a visual scene displaying a pair of objects. They performed left key presses when the target was on the left and right key presses when the target was on the right. Target handle orientation and response side could be compatible or incompatible. Critically, target and distractor objects had similar or dissimilar handle affordances, with handles oriented for left- or right-hand grasps. Results from the two experiments showed slower response times when target and distractor objects had similar handle affordances in comparison to dissimilar affordances, when participants performed right hand responses and when target orientation and response were compatible. Thus, affordance similarity between objects may interfere rather than facilitate object processing and slow down target selection. These findings are in line with models of affordance and object selection assuming automatic inhibition of distractors' affordances for appropriate object interaction.

## Introduction

When observers perceive their environment, they also perceive the action possibilities that the environment offers, i.e. "affordances". The concept of affordance, first defined by Gibson [1] has been later-on applied to objects such as tools in a cognitive approach. Ellis and Tucker [2]

**Data Availability Statement:** Raw data and analyses codes are available at https://doi.org/10.5281/zenodo.7648888.

**Funding:** This work benefitted from a PhD fellowship from the University of Lille awarded to the first author.

**Competing interests:** The authors have declared that no competing interests exist.

proposed the term "micro-affordances" to refer to the different action components that a given object potentiates. Depending on the size, position in space and handle orientation of the object, different micro-affordances may be activated. Micro-affordances correspond to the hand effector (left- or right-hand grasp), the wrist orientation (horizontal or vertical grasp) and the grasp size (power or precision grasp) that would be best suited to grasp the object [3, 4]. The existence of micro-affordances, and in particular those related to the hand effector, has been classically demonstrated using the Stimulus-Response Compatibility (SRC) paradigm. In this paradigm, graspable objects (e.g., a pan, a mug) are presented with their handles oriented to the left or to the right. Participants are then asked to perform a categorization task on the objects using left or right key press responses that are compatible or incompatible with the handle orientation of the visual object presented. A compatibility effect has been reported, with shorter response times when both the handle orientation of the object and the response hand are compatible in comparison to incompatible [3, 5]. Similar results have been observed for grasp size compatible with object size [5]. Overall, the compatibility effect between the visual properties of the object and the motor properties of the response may be taken as evidence of the activation of micro-affordances from visual objects.

Yet alternative explanations of compatibility effects have been proposed, relying on the compatibility between abstract codes associated to the stimulus and the response [6–8]. The stimulus could be coded as abstractly [small] or [large] in opposition to evoking precision or power grasps. Similarly, the response could also be considered as a [small] or [large] response independently of the type of grasp. A compatibility effect would arise when the abstract codes of both the stimulus and response match. However, recent findings indicated that the combination of specific types of task (e.g., a task relevant for action) and response (e.g., reach-and-grasp response) could favour the activation of action components [9]. The affordance hypothesis is also legitimated by neurophysiological studies highlighting an activity of the motor system during the perception of manipulable objects, independently of the SRC paradigm [10–13]. In consequence, although abstract coding may be frequently at play in compatibility effects, it does not completely rule out the existence of affordance activation in some specific situations.

Furthermore, affordance activation in stimulus-response compatibility paradigms has been classically studied in very simple and artificial situations where the object stimulus is presented in isolation and may be compatible with one single motor response. This provides a very restrictive view of the potential scope of affordance effects. It is now critical to consider micro-affordances in the context of more complex and realistic perceptual situations. First, a given object does not necessarily evoke only one single micro-affordance. If one considers for example a calculator, one would typically use a power grasp to grasp the calculator, but a poke on the keys to use it. Previous studies showed that in a neutral context and when the object was presented with no prior action intention, both grasp affordances could be activated from a single object [14, 15] and compete with one another. This competition has a cost: initiation of an action as well as perceptual judgments towards an object are slower when the object activates distinct affordances as compared to similar affordances [16–18]. Neurophysiological studies further showed that the competition between distinct affordances during single object perception extinguished motor resonance effects visible in Mu rhythm desynchronization [16, 19]. Overall, recent behavioral and neurophysiological studies demonstrated that the perception of a single manipulable object can be affected by the diversity of the affordances activated, even in the absence of action intention. Therefore, the impact of multiple affordances on single object perception may be related to the involvement of the motor system in object perceptual processing. Second, natural perceptual scenes are rarely composed of isolated objects but usually feature multiple objects. Without considering the evocation of affordances, the influence

of distractors on target identification has been investigated for target and distractors sharing similar or dissimilar visual properties in classical flanker tasks [20]. Authors usually highlight slower response times to identify the target when target and distractors shared dissimilar visual properties in comparison to similar visual properties. One may then wonder if the cost found for target identification when distractors shared dissimilar visual properties with the target may be also found when target and distractors evoke dissimilar motor properties or affordances.

Previous research on affordance activation in multi-object perceptual situations is very limited. In multi-object situations, the other objects in the scene provide a context and potentiate the way we perceive a given object [21–23]. For instance, when an object-tool pair is presented within a visual scene, each object in the pair does not only activate the action possibilities it would typically afford when presented alone but also those associated with the common or uncommon use of the tool in conjunction with the specific object from the pair. This is the case when a knife is situated near a screw, it may suggest the action of "screwing" rather than the typical action of "cutting." However, it is far from clear whether competition phenomena arise from distinct affordances evoked by multiple objects. The presence of distractor affordances raises several important issues. First, does an object still activate affordances when presented among other objects? A study of Derbyshire et al. [24] provided a first insight into affordance perception in multi-object scenes. Scenes of four natural and artefact objects that evoked power or precision grasps were presented. An arrow appeared and pointed toward a target object, the other three objects were considered distractors. Participants had to determine if the target was a natural or artefact object by performing power and precision grasp responses compatible or incompatible with the affordance evoked by the target. They found a compatibility effect between target and response with faster response times when target and response were compatible in comparison to incompatible. Results suggest that in the presence of distractor objects, the affordance of a target object is still perceived and activated. However, this study did not investigate the influence of the affordances of distractor objects on the processing of the target object. How does distractor affordances impact target processing and selection? On the one hand, one may expect similar affordance competition mechanisms in single-object and multi-object scenes. Competition between affordances evoked by the different objects of the scene would entail a perceptual processing cost, with slower processing of a target object when distractor objects have dissimilar affordances, in comparison to similar affordances. The neurobiological model of action selection proposed by Cisek [25] is compatible with this prediction. When several affordances are simultaneously available in the environment, observers would first activate all the different possible affordances in parallel. Information would then be accumulated from various sources (e.g., sensory information about possible targets, motor information about potential reaching movements, cognitive information about goals and expected utility of actions. . .) in order to bias the competition and select the most relevant affordance to interact with the target object. In this framework, one may thus expect distractors with dissimilar affordances to interfere more with the processing of the target object than distractors with similar affordances. The processing of distractors with dissimilar affordances would cumulate the duration of two processes: affordance activation and affordance selection.

On the other hand, a few studies addressing the issue of affordance activation in the context of object selection showed diverging results. For instance, Pavese and Buxbaum [26] investigated the competition between object grasps evoked by multiple handled objects (i.e., cups). One cup was the target that could be presented with or without a distractor cup. Cups were presented in different configurations 1) with target and distractor handles oriented to the left or to the right (compatible with a left or right grasp, leading to similar or dissimilar handle

affordances), 2) with target and distractor handles visible or invisible. The target object would then change color and participants had to perform a response toward the target object. Different types of responses were compared: reach and grasp responses (i.e., reach and grasp the handle of the target object), reach and point responses (i.e., reach and point the handle of the target object) and button press responses (i.e., press the upper button if the target handle is oriented to the right and the bottom button if it is oriented to the left and conversely). Interference from distractors was measured, corresponding to the difference between "target alone" and "target and distractor" conditions. Overall, interference from distractor objects with visible handles on the processing of target objects was observed. For action relevant responses (i.e., reach-and-point and reach-and-grasp responses), there was more interference when distractors had similar than dissimilar affordances. In contrast, for action irrelevant responses (i.e., button presses), there was more interference when distractors had dissimilar than similar affordances, which seems to be related to the visual salience of the handles in the conditions of similar affordances. These results are supported by results reported by Bub et al. [9] and Ellis et al. [27]. Ellis et al. [27] showed that when observers had to select a target object (i.e., a cube or a cylinder) among other distractor objects (i.e., cubes or cylinders) by performing grasp gestures, initiation times were slower when both target and distractors evoked similar in comparison to dissimilar affordances. In these studies, as in the action-relevant tasks of Pavese and Buxbaum's study, distractors with similar affordances interfered more with the processing of the target object than distractors with dissimilar affordances. Yet results remain very limited, as to our knowledge only these few studies have investigated the effect of distractors' affordances on the processing of a target object. In addition, the use of impoverished stimuli such as cubes or cylinders [27] may have limited the potential activations of object affordances.

The existence of a cost of affordance similarity in multi-object situations would be compatible with the automatic inhibition hypothesis proposed by two models in the context of object selection [28, 29]. When observers must select a target object among distractors, the distractor affordances would be automatically inhibited to allow the most efficient interaction with the target object. When both target and distractor evoke similar affordances, the inhibition of the distractor affordance would also result in the inhibition of the target affordance, as both affordances are similar. The processing and selection of the target object would then be slowed down. This cost would not be found when target and distractor have dissimilar affordances. Only the distractor affordance should be inhibited, and the target object would be processed and selected directly. Predictions of the inhibition hypothesis have been supported by some empirical data [30–33]. In one study, Vainio et al. [31] presented to participants an object with its handle oriented for a right or left-hand grasp. The object served as a prime of a target line or target arrow oriented to the left or to the right. In a go-no go task, participants had to refrain to answer when the target was a line but had to determine the direction of the arrow by pressing with their left thumb if the arrow pointed to the left and with their right thumb if the arrow pointed to the right. Overall, results showed that participants took longer to judge the direction of the target arrow when it was presented in an orientation similar to the handle of the non-target object prime, as compared to dissimilar. In addition, in no-go trials, participants tended to incorrectly respond to the target more when target and prime objects were dissimilarly oriented, as compared to similarly oriented. These results are in line with the predictions of the inhibition hypothesis, as distractor objects with orientation properties similar to the target and response seem to interfere with target processing, more than distractor evoking dissimilar properties. Furthermore, errors in no-go trials provided additional support in favor of a mechanism based on affordance inhibition: participants had more difficulty to refrain from responding when distractor affordances were dissimilar. However, although the few empirical

data presented are consistent with inhibition hypothesis, the different predictions still need to be investigated with scenes of familiar objects.

The aim of the present study was then to determine the impact of distractor affordances on the processing of a target object in situation involving several familiar graspable objects (i.e., kitchen utensils and tools) investigating to what extent affordances of distractor objects compete with the affordance of a target object. For this purpose, we conducted two experiments where we presented scenes of two objects with handles, one object being the target, the other one the distractor. Affordances considered in the present study were handle affordances and corresponded to the hand effector evoked by the orientation of the object handle. Handles were oriented to the left or to the right evoking left- or right-hand handle affordances. Both objects could evoke similar handle affordances (e.g., both objects oriented for a right-hand grasp) or dissimilar affordances (e.g., one object oriented for a right-hand grasp and the other for a left-hand grasp). Responses were performed with left and right key presses that could be compatible or incompatible with the target object handle affordance. We first expected a compatibility effect with participants being faster when the target object are compatible with the response performed in comparison to when target object and response are incompatible. Following the predictions of Caligiore et al. [28] and Vainio and Ellis [29], we further expected a greater interference from distractors with similar compared to dissimilar handle affordances on this compatibility effect: distractor objects evoking handle affordances similar to the target should slow down target processing. Distractors with similar affordances should entail an interference effect when target and response are compatible, as the action performed toward the target is relevant. As in incompatible conditions the response toward the target is irrelevant, the affordance of the distractor object should have a lesser impact.

## Experiment 1

### Methodology

**Participants.**    Participants were recruited on the Prolific platform (www.prolific.co). One hundred and forty-six participants (57 women) between 18 and 40 years old were recruited ($M = 27.33$, $SD = 5.49$). Inclusion criteria included being right-handed, speaking French and living in France. Object affordances were considered from the perspective of right-handers and French language or culture was selected to ensure familiarity with the object exemplars used. Participants were informed about the study by receiving an automatic email from Prolific with the experiment details if they met the experiment inclusion criteria. They were aware that the study focused on visual perception of objects and aimed to understand how we categorize object among distractors. When clicking on the link to the study, they were again informed about the objective of the study. They were also informed about the task, namely that they will see scenes of two objects and will have to determine if objects are kitchen utensils or tools. They were then reminded of the potential risks and benefits and their rights as human participants and written consent was obtained electronically. A random anonymous code was attributed, and handedness was verified according to an online adaptation of the Oldfield Edinburgh test [34]. Then, the program continued to the experiment with a second reminder of the instructions. They were paid £7.52 per hour to compensate for their participation in the study. The recruitment and the testing of the participants were in conformity with the Helsinki declaration of 1964. The protocol was approved by the Ethical Committee of the University of Lille.

**Justification of the number of participants.**    An a priori power analysis was conducted with R software using the function pwr.t.test of the pwr package (v1.3–0; [35]). As effect sizes could not be directly anticipated from previous studies, we reasonably expected a relatively

**Target = Whisk**

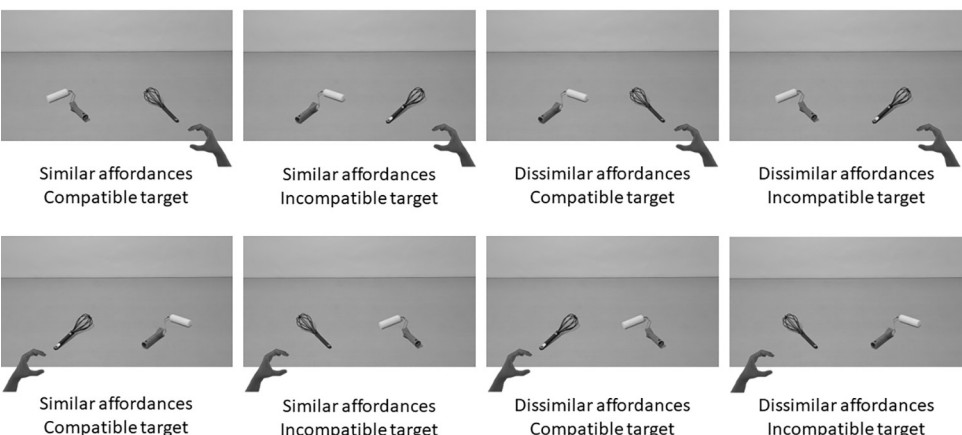

**Fig 1. Example of an object pair in the 8 conditions of presentation.** Objects are presented on a table at equal distances from the centre. Both objects could be similarly oriented or not: evoking similar or dissimilar handle affordances. The target (here the whisk) could be oriented to the left or to the right to be compatible or incompatible with the response hand (left- or right-hand responses).

small effect size (Cohen's d between 0.2 and 0.3) for the effect of similarity between target and distractor affordances on the handle compatibility effect in the present protocol. To guarantee a sufficient statistical power for a two-tails hypothesis ($\beta = 0.2$; power ($1- \beta$) = 0.80; $\alpha = 0.05$), about 120 participants were anticipated.

**Stimuli.** Stimuli were 96 photographs of scenes involving two manipulable objects displayed on a table. Pictures were presented in their black and white version to avoid any salience effect due to colors. Both objects were presented at an equal distance from the center of the picture, one on the left and one on the right, at fixed locations. Each object was positioned based on the middle of the horizontal axis of the whole object. As picture dimensions were 1920x1080 pixels, objects were placed on the left and on the right at 370 pixels from the midline. Objects appeared on the table in the lower part of the screen at 290 pixels from the bottom of the picture. The objects were 12 tools and 12 kitchen utensils with handles (e.g., a paint roller or a whisk) organized in fixed pairs with one kitchen utensil and one tool within each pair (i.e., the whisk was always presented with the paint roller, Fig 1). Each pair of objects was presented 8 times in different configurations: objects could be oriented with their handles to the left or to the right, evoking a left- or right-hand grasp, and with the tool on the left and the kitchen utensil on the right or conversely, for a total of 12 set * 8 configurations = 96 stimuli. A set of eight additional stimuli were used as examples. The complete list of stimuli can be found in S1 Appendix.

**Response modalities.** Participants had to perform the task by pressing keys on the keyboard of a computer. Participants saw scenes of two objects and had to select the target, namely the kitchen utensil or the tool, the task being randomly assigned by the program to each participant. Two responses were possible: when the target was the object on the left, they performed a left response with simultaneous presses of both "e" and "c" keys. When the target was on the right, they performed a right response with simultaneous presses of both "i" and "n" keys. Participants pressed the keys using their index finger and thumbs. Those response modalities were chosen so the hand posture of the participants would mimic a grasp (Fig 2). Even though the grasp aperture of the response was not necessarily tuned to the object, the size of the grasp was not critical to our design as we manipulated the affordance compatibility between handle

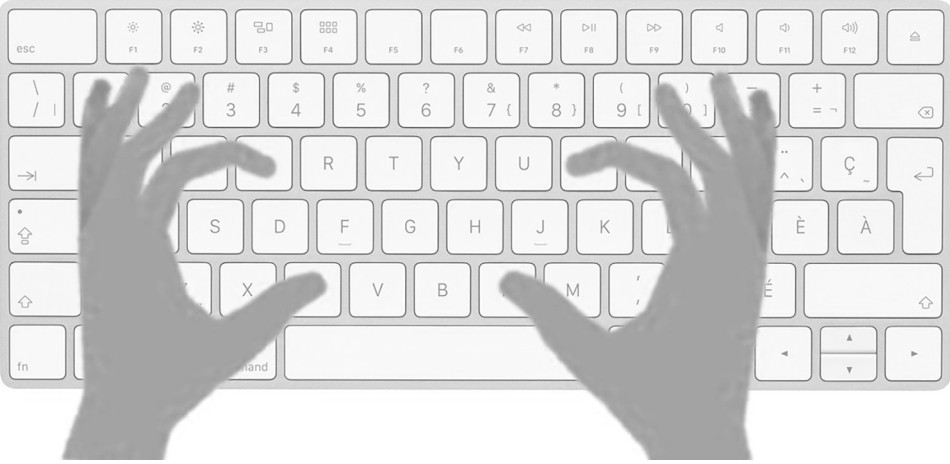

**Fig 2. Schematic representation of the response modalities.** Participants had to determine on which side appeared the tool/the kitchen utensil (task randomly assigned between participants) by to pressing keys located on the left and right of the keyboard. They simultaneously pressed the "e" and "c" keys when the target was on the left and "i" and "n" keys when the target was on the right. Participants had to press the keys with their index finger and thumb so the hand posture would mimic a grasp.

left/right orientation and left/right response and not between the size of the object and the size of the grasp response.

**Procedure.** The experiment was built on Psychopy software [36] and conducted online using Pavlovia, Pyschopy online extension. Participants received the information about the experiment by email. The experiment was built on Pavlovia to control participants are using computers to launch the task. When launching the experiment, a random anonymous code (14 random letters and numbers) was attributed to each participant, and they received the written instructions a second time. Participants had to consent to participate to the study by pressing the space bar to continue. They were asked to seat in front of their computer screen at approximately 40 cm and to be aligned with the middle of the screen. The experiment was built to accommodate all screen sizes possible, so despite possible variations in screen size the relative dimensions and positions of objects in the scene was constant. Participants were then randomly assigned to one of the two task versions, half of the participants had to select the target as the kitchen utensil and the other half as the tool. After completing an online version of the Oldfield questionnaire, participants performed a short training session followed by the experimental session. At the beginning of each trial, an empty scene appeared for 500 ms, followed by a fixation cross during 500 ms. Then, the two objects were displayed on the screen and participants had to select the target object. For half of the participants, the target object was always the kitchen utensil and for the other half it was always the tool. They had to answer as accurately and quickly as possible on the keyboard by pressing simultaneously the "e" and "c" keys if the target object was on the left or the "i" and "n" keys if object was on the right. Participants had as much time as needed to respond to the task. After responding, the scene disappeared, and another empty scene appeared for the next trial (Fig 3). Timing of the trial procedure was pre-tested in the laboratory before conducting the online experiments. The experiment consisted of 96 trials of 12 pairs of objects presented randomly in 8 different conditions. First, object handle affordances could be similar or dissimilar. Objects could be similarly oriented for a right-hand grasp or similarly oriented for a left-hand grasp (similar condition) or not similarly oriented, one for a right-hand grasp and one for a left-hand grasp (dissimilar condition). Second, the orientation of the target object and response could be compatible or

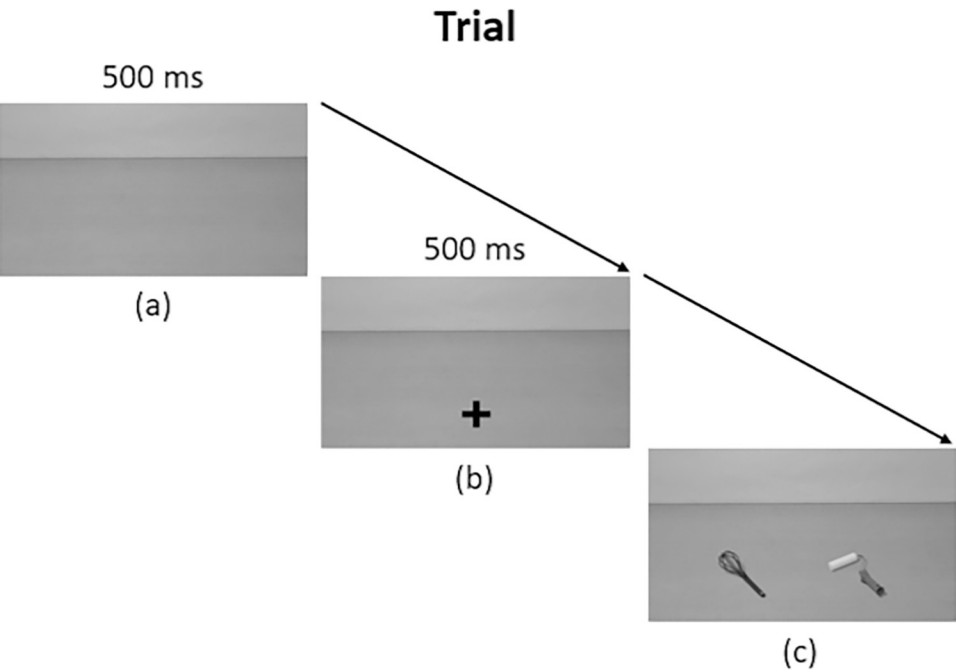

**Fig 3. Schematic representation of a trial.** (a) An empty scene was displayed during 500 ms. (b) A fixation cross appeared for 500 ms and participants were asked to look at the cross. (c) The objects were displayed on the scene and participants were asked to select the target object, which was instructed as either the kitchen utensil or the tool (randomly assigned between participants), by pressing keys on the same side of the keyboard. The next trial began after participants responded.

incompatible. The target object could be presented on the right and oriented for a right-hand grasp or presented on the left and oriented for a left-hand grasp (compatible condition) or presented on the right and oriented for a left-hand grasp or presented on the left and oriented for a right-hand grasp (incompatible condition). The response hand always corresponded to the location of the target, as participants had to press on the side of the target. A break was proposed halfway through the experiment. When the task ended, a written debriefing of the experiment and the contact details of the scientific coordinators were displayed. The median duration of the experiment was 9.31 minutes.

## Results

**Data preprocessing.** Data pre-processing was conducted using R software with the packages plyr (v1.8.6; [37]), tidyverse (v1.3.1; [38]), lme4 (v1.1–27.1; [39]), afex (v1.0–1; [40]), broom.mixed (v0.2.7, [41]) and emmeans (v1.7.2; [42]).

First, the Oldfield questionnaire was analyzed, and participants who were not confirmed strongly right-handed by the questionnaire (cut-off = 50; n = 16) were excluded. Then, errors in the data set were identified. In addition to responses on the side of the incorrect object, we categorized as errors responses made with a difference superior to 100 ms between the two key presses in order to ensure that participants responded with a grasp hand posture. We considered 100 ms as a minimum for a simple reaction time, following Woods et al. [43], and thus considered two key presses separated by more than 100 ms as two different actions. We also considered as errors presses that were made with only one out of the two key presses required or responses with three key presses. Following this procedure, 12% of error trials were excluded. Response times (RT), considered as the mean response times of the two key presses,

were preprocessed on correct trials. A global trim was undergone by excluding RT inferior to 200 ms and RT superior to 4000 ms, to remove any aberrant responses. We then excluded RTs superior to 2.5 standard deviations from the mean RT of each participant in each condition (affordance similarity x target and response compatibility x task version). With this trimming procedure, 3.14% of RT were excluded. Finally, outlier participants after the error and RT trimming procedure were excluded. Eight participants with accuracy above or below 2.5 standard deviation from the mean accuracy or the mean RT of the group were excluded. Note that for all experiments, the same patterns of results were observed with alternative RT pre-processing procedures such as trimming based on Median Absolute Deviation (MAD) or analyses on log-transformed data. Analyses on non-transformed RTs after trimming based on mean/SD was chosen as the best compromise to simultaneously consider the skewness of the RT distribution, the interpretation of the model estimates and the proportion of excluded trials. Finally, three participants with more than 35% of missing trials were excluded. One-hundred and twenty-two participants were kept for further analyses, 62 in the task version where the target was the kitchen utensil and 60 in the version where the target was the tool. Since accuracy was not at ceiling ($M = 0.87$, $SD = 0.32$), accuracy data were also analyzed to verify the absence of speed-accuracy trade-off.

**Data analysis.** Logistic mixed-effect models were used to analyze accuracy and linear mixed-effect models were undergone to analyze RTs. Analyses were conducted with the lmer function of the lme4 package (v1.1–12; [39]) in R software (version 2021.09.1). In our models the fixed effect factors included i) the Similarity of handle affordances between target and distractor (similar and non-similar), ii) the Compatibility between target orientation and response (compatible, incompatible) and iii) the Response Hand corresponding to the target location (left, right). We included the maximal random effect structure supported by the data and the model. Random effects were considered for participants, nested in Task version, as well as for items. To choose random structures, we followed the guidelines proposed by Barr et al. and Bates et al. [44, 45]. We first built our model with the maximal random structure possible. If the model did not converge, we reduced the random structure. To do so, we ran Principal Component Analyses to estimate for each intercept and slope of our model the part of variance explained. We used the rePCA function from the lme4 package (v1.1–27.1; [39]). We kept the intercepts and slopes that explained the biggest part of variance and removed the ones explaining only a small percentage of variance. This process was repeated until the model converged. The complete mixed-effect model structures can be found in S2 Appendix. Effect sizes were computed as Wesftall's d, an alternative to Cohen's d suitable for linear mixed model effects [46, 47]. Westfall's d measures were computed with the eff_size function of the emmeans package [42].

We first expected a significant main effect of Compatibility which would reflect an overall compatibility effect. Response times should be shorter when target object and response are compatible (i.e., when the target handle is oriented to the right and a right keypress response and target handle oriented to the left and a left keypress response) in comparison to incompatible (i.e., when the target handle is oriented to the right and a left keypress response and conversely). Secondly, we specifically expected an interaction between Similarity and Compatibility, which would reflect an influence of distractor objects on the compatibility effect. Following the inhibition hypothesis of Caligiore et al. [28] and Vainio and Ellis [29], We anticipated that distractors with handle affordances similar to the target objects would interfere with the processing of the target object. In the compatible condition, response times should be faster for target and distractor with similar handle affordances in comparison to dissimilar affordances.

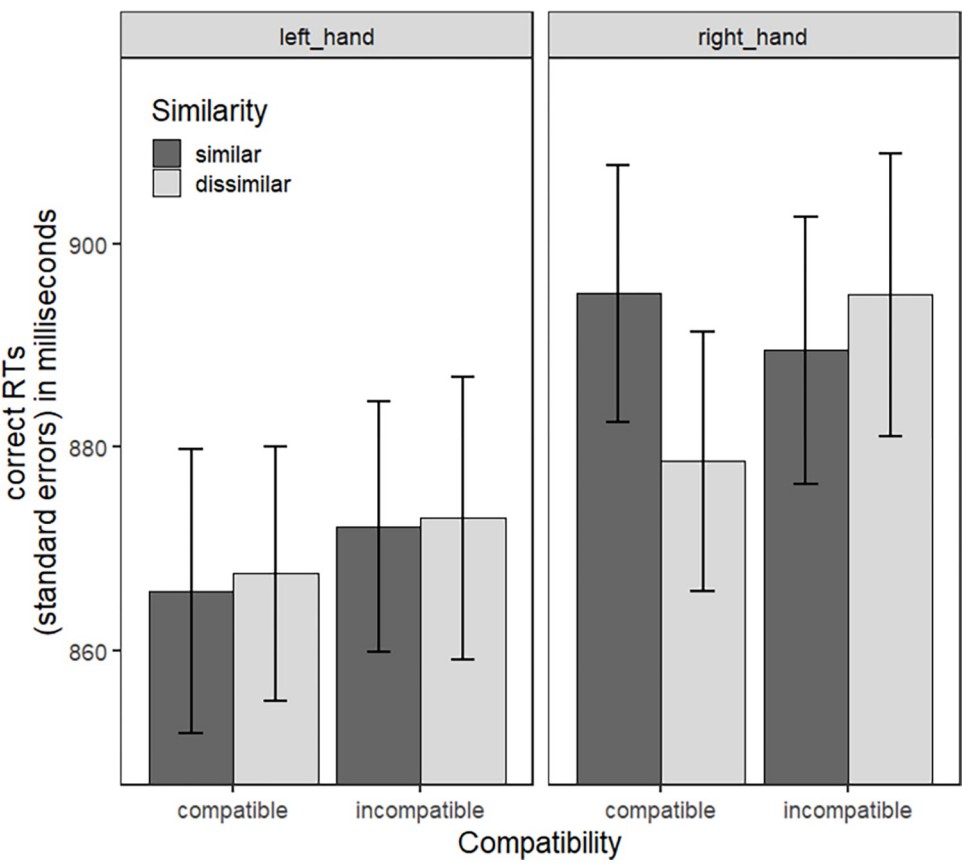

**Fig 4. Mean RTs as a function of Similarity, Compatibility and Response Hand for Experiment 1.** Error bars correspond to the standard errors for participants. They are informative of the variability between participants in each condition but not of the variability of the within-subject effects that we statistically evaluated.

**Accuracy.** No significant effects were found for the logistic mixed- effect model on accuracy. The accuracy was then not further analyzed.

**Response times (RTs).** The expected interaction between Similarity and Compatibility was not significant (estimate = 4.164, $t$ = 1.005, SE = 4.144, $p$ = 0.314, Westfall's d = 0.016). Interestingly, even if the effect did not reach significance, there was a marginal three-way interaction between Similarity Compatibility, and Response Hand (Fig 4; estimate = -10.150, $t$ = -1;732, SE = 5.861, $p$ = 0.083, Westfall's d = 0.040).

Regarding main effects, we did not find any significant main effect of Similarity (estimate = -1.472, $t$ = -0.473, SE = 3.112, $p$ = 0.637, Westfall's d = 0.008) or Compatibility (estimate = 3.089, $t$ = 0.917, SE = 3.368, $p$ = 0.360, Westfall's d = 0.017). However, a significant main effect of the response hand/target location was found (estimate = -15.397, $t$ = -3.598, SE = 4.279, $p$ < 0.001, Westfall's d = 0.085), with overall faster RTs for left responses ($M$ = 869.70, $SD$ = 132.88) than right responses ($M$ = 890.45, $SD$ = 132.47).

The descriptive statistics including mean RTs and standard deviations for the Similarity x Compatibility x Response Hand conditions are provided in S3 Appendix.

## Discussion

This experiment aimed at investigating whether the similarity of handle affordances between target and distractor could influence target object selection. The hypothesis was that when

target orientation and response are compatible, target selection should be facilitated when the handle affordances of target and distractor are dissimilar, in comparison to when they are similar. Results did not support the hypothesis. However, results suggested that the response hand associated with the left/right location of the handle of the target might influence target selection. The interaction was not particularly expected a priori, but could be relevant from an affordance perspective, especially considering the possible influence of the response hand on handle affordance similarity and compatibility effects.

Participants were faster to perform left responses than right responses. The direction of the effect might be surprising if we consider it from the perspective of the hand used to respond. One may have expected faster response times for right hand responses in comparison to left hand responses in right-handed participants [48]. In addition, the possible modulation of the critical interaction between affordance similarity and target-response compatibility was also unpredicted. An alternative interpretation might relate to differences in processing the left and right visual fields, rather than differences in performing left and right-hand responses, since the two variables are confounded here. Participants may allocate more attention to the left than the right part of the scene. As instructions required to select the target object based on object category and target and distractor objects always belonged to different categories, one possible strategy to complete the task could have been to look at one object only. If, based on reading experience, participants systematically started the exploration by looking at the left part of the scene, they could directly respond on the left when the target appeared on the left and indirectly respond on the right by deducting that the target was on the right when the distractor from the alternative category appeared on the left. In that case scenario, the disadvantage of right responses would be a consequence of a predominant exploration of the left visual field and simply reflect the time needed to infer target location when it was displayed on the unattended side. We thus conducted a second experiment following the same procedure as Experiment 1 in order to rule out this possible attentional bias. We aimed at evaluating whether left hand responses remained faster than right hand responses in this paradigm when the procedure required participants to allocate the same level of attention to both sides of the scene. To do so, "catch-trials" displaying two identical objects were added, for which participants had to refrain from responding.

## Experiment 2

### Methodology

**Participants.** Participants were recruited on the Prolific platform (www.prolific.co) following the same recruitment procedure as Experiment 1. The same inclusion criteria were used, but participants who participated in Experiment 1 could not participate in Experiment 2. One hundred and thirty-seven participants (68 women) between 18 and 40 years old were recruited ($M = 26.94$, $SD = 5.34$). They were paid £ 7.50 per hour to compensate for their participation in the experiment.

**Stimuli.** The stimuli were the same as the ones used in Experiment 1. In addition to the 96 scenes, 24 additional scenes were added for the design of catch-trials. Those new scenes were pairs of identical objects: both objects could be the same kitchen utensil or the same tool, with their handles oriented to the left or to the right.

**Response modalities.** The response modalities were the same as in Experiment 1. In addition to the two previous responses, participants were also asked to refrain from responding on catch-trials. Thus, participants could perform left key presses, right key presses, or no key presses.

**Procedure.**   The procedure was similar to that of Experiment 1, with the exception of the additional presentation of scenes involving two identical objects. When pairs of identical objects were displayed (12 pairs of kitchen utensils for participants instructed to select kitchen utensils and 12 pairs of tools for participants instructed to select tools), participants were asked to refrain from responding. Those catch-trials were displayed to force participants to pay equal attention to both objects in the scene. The median duration of the experiment was 10.41 minutes.

## Results

**Data preprocessing.**   Data pre-processing was conducted in R software using the same procedure as Experiment 1.

First, participants who were not confirmed as right-handed [34] were excluded (cut-off = 50; n = 14). In addition to the prior pre-processing steps, participants in the 2nd experiment who responded on more than 20% of the catch trials (5/25) were also excluded. Then, errors in the data set were identified. With this trimming procedure, 10.4% of error trials were excluded. Response times (RT) were preprocessed on correct trials. 3.12% of RT were excluded. After the error and RT trimming procedure, eight outlier participants with accuracy above or below 2.5 standard deviation from the mean accuracy or the mean RT were excluded. Finally, six participants with more than 35% of missing trials were excluded. One hundred and seven participants were kept for further analyses, 56 had to select the kitchen utensils and 51 had to select the tools. Since accuracy was not at ceiling ($M = 0.90$, $SD = 0.29$), accuracy data were also analyzed to verify the absence of speed-accuracy trade-off.

**Data analysis.**   Logistic mixed-effect models were used to analyze accuracy and linear mixed models were conducted to analyze response times. The fixed effects were the same as in Experiment 1: Similarity, Compatibility, and Response Hand. Random structure selection followed the same procedure as for Experiment 1. The complete structure of mixed- effect models used in Experiment 2 can be found in S2 Appendix. Effect sizes were computed as Westfall's d.

**Accuracy.**   No significant effects were found in logistic mixed- effect model on accuracy data. The accuracy was then not further analyzed.

**Response times.**   As for Experiment 1, the interaction between Similarity and Compatibility (estimate = 4.775, $t = 1.060$, SE = 4.505, $p = 0.289$, Westfall's d = 0.018) did not reach significance. The three-way interaction between Similarity, Compatibility and Response Hand (Fig 5; estimate = -9.486, $t = -1.488$, SE = 6.372, $p = 0.136$, Westfall's d = 0.038) was not significant either.

As for the main effects, neither the main effect of Similarity (estimate = -0.614, $t = -0.193$, SE = 3.186, $p = 0.847$, Westfall's d = 0.002) nor the main effect of Compatibility (estimate = 2.327, $t = 0.731$, SE = 3.186, $p = 0.465$, Westfall's d = 0.016) were significant. We only found a significant main effect of Response Hand (estimate = -13.977, $t = -3.437$, SE = 4.067, $p < 0.001$, Westfall's d = 0.052), with again shorter RTs for left ($M = 886.20$, $SD = 139.26$) than right responses ($M = 906.50$, $SD = 129.67$).

The descriptive statistics including mean RT and standard deviations for the Similarity x Compatibility x Response Hand condition are provided in S3 Appendix.

## Discussion

Experiment 2 aimed at investigating whether the influence of the response hand on the pattern of RT for target selection could be due to an attention bias to the left coupled with a strategy based on single object processing, with greater if not exclusive attention allocated to objects presented on the left part of the scene compared to objects on the right part of the scene. As in

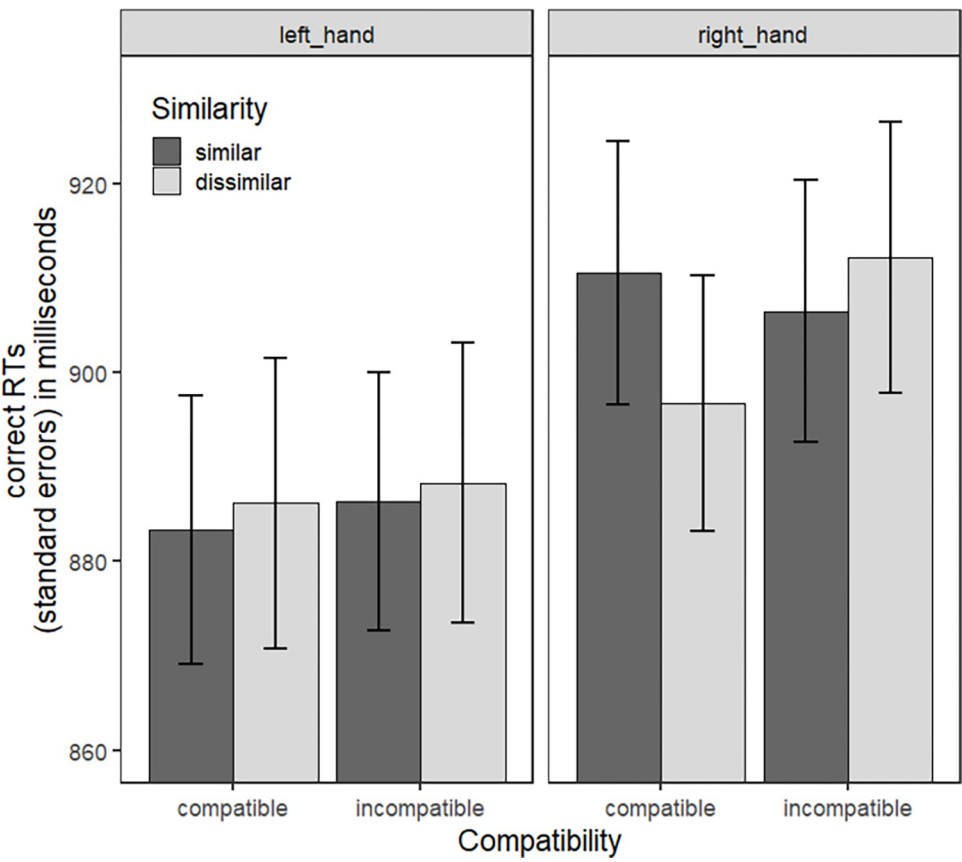

**Fig 5. Mean RTs as a function of Similarity, Compatibility and Response Hand for Experiment 2.** Error bars correspond to the standard errors for participants. They are informative of the variability between participants in each condition but not of the variability of the within-subject effects that we statistically evaluated.

Experiment 1, we observed a significant effect of the response hand, with faster RT for left than right responses, despite the presence of catch trials that required looking at both objects before making a response. No other significant effects were found. Therefore, we are relatively confident that the facilitation for left hand responses does not arise from a visuo-attentional factor. Alternatively, we believe that the response hand may in fact influence the speed of target selection. This hypothesis would also explain possible differences in handle affordance similarity and compatibility effects depending on whether participants respond with their dominant hand or not.

## Between-experiment analysis

In order to more directly quantify the influence of the experiment version on the pattern of results observed, we finally conducted a between-experiment analysis. This analysis further gave us the opportunity to investigate a potential modulation of affordance similarity and compatibility effects by the response hand in a larger sample of participants.

### Results

The analysis involved 229 participants, 118 who had to select the kitchen utensil and 111 who had to select the tool.

Linear mixed models were used to analyze trimmed correct RTs from both experiments. They were conducted with the lmer function of the lme4 package (v1.1–12; [39]) with the R software. Mixed model structure was chosen based on the same procedure as explained before. In addition, the experiment version was added in the random structure. Effects size were computed as Westfall's d.

**Response times.** Similarity, Compatibility, Response Hand and Experiment Version were included in the model as fixed effects. The random structure included random effects for participants nested in Task version and Experiment. Items were also included in the random structure. The final model structure can be found in S2 Appendix.

First, the Experiment Version was not interacting significantly with our effects of interest: neither the four-way interaction between Similarity, Compatibility, Response Hand and Experiment Version (estimate = 0.636, $t$ = 0.104, SE = 6.130, $p$ = 0.917, Westfall's d = 0.002) nor the three-way interaction between Similarity, Compatibility and Experiment were significant (estimate = 0.462, $t$ = 0.107, SE = 4.333, $p$ = 0.915, Westfall's d = 0.001). In addition, no main effect of Experiment Version was found (estimate = 12.256, $t$ = 1.011, SE = 12.118, $p$ = 0.312, Westfall's d = 0.055).

The three-way interaction of interest between Similarity, Compatibility and Response Hand was significant (Fig 6; estimate = -9.877, $t$ = -2.279, SE = 4.334, $p$ = 0.022, Westfall's d = 0.039). In the compatible condition, paired comparisons showed that RTs were 16 ms longer when target and distractor handle affordances were similar ($M$ = 902.416, SD = 142.230)

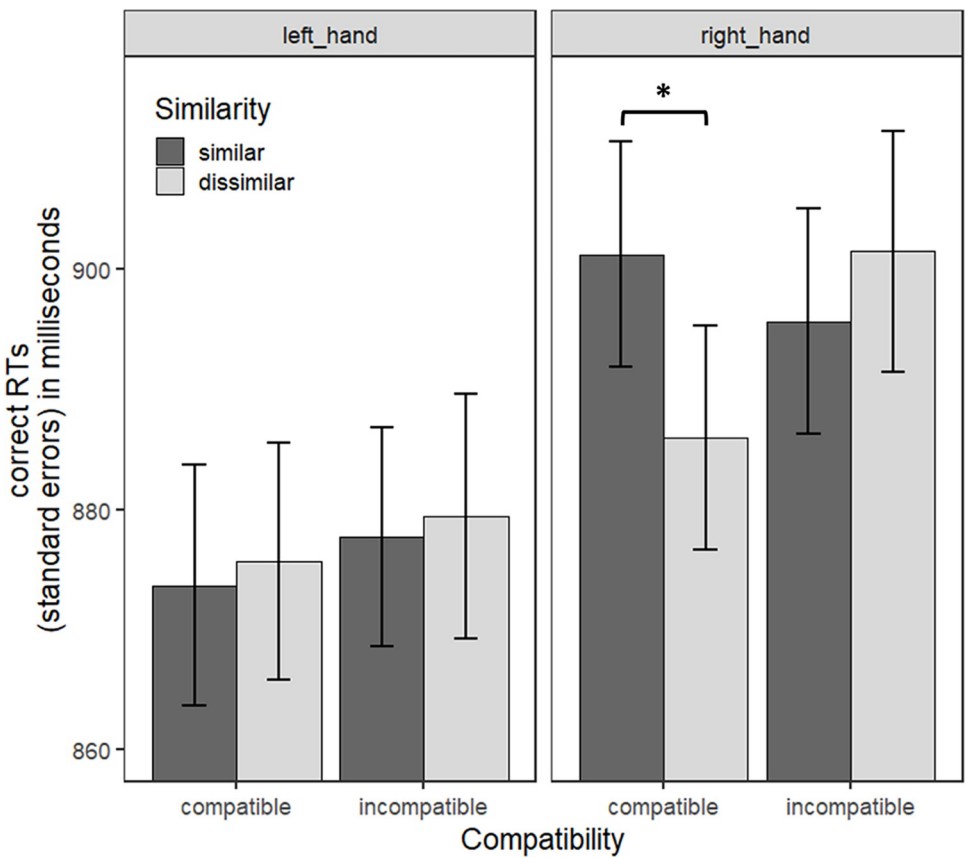

**Fig 6. Mean RTs as a function of Similarity, Compatibility and Response Hand of data from Experiment 1 and 2.**
* $p$ < .05. Error bars correspond to the standard errors for participants.

compared to dissimilar (*M* = 887.158, *SD* = 141.299), when participant responded with their right hand (estimate = 16.520, *z* = 2.695, SE = 6.130, *p* < 0.01, Westfall's d = 0.064). There was no such difference for left-hand responses (estimate = -4.224, *z* = -0.690, SE = 6.120, *p* = 0.490, Westfall's d = 0.016). In addition, when target and response were incompatible, this interference effect for similar compared to dissimilar handle affordances when responding with the right hand was not visible (estimate = -6.639, *z* = -1.081, SE = 6.140, *p* = 0.279, Westfall's d = 0.002). We verified the interaction between Similarity and Compatibility for each Response Hand with additional sub-models (see S2 Appendix). Results confirmed that the Similarity x Compatibility interaction was significant for the right hand (estimate = 11.679, *t* = 2.709, SE = 4.311, *p* < .01, Westfall's d = 0.046) but not for the left hand (estimate = -2.497, *t* = -0.578, SE = 4.318, *p* = 0.563, Westfall's d = 0.010).

No significant main effects were found for Similarity (estimate = -1.098, *t* = -0.507, SE = 2.167, *p* = 0.612, Westfall's d = 0.006) and Compatibility (estimate = 2.570, *t* = 1.047, SE = 2.454, *p* = 0.296, Westfall's d = 0.015). There was however a significant main effect of the Response Hand (estimate = -14.658, *t* = -4.925, SE = 2.976, *p* < 0.001, Westfall's d = 0.082). The descriptive statistics including mean and standard deviation of the reaction times for Similarity x Compatibility x Response Hand for the combined experiments are provided in S3 Appendix.

**Temporal dynamics of distractor affordance effects.**   Support for inhibition processes has been often sought in the temporal dynamics of stimulus-response compatibility effects [49]. Delta plots displaying the RT difference between compatible and incompatible conditions as a function of response time distribution are typically used to this aim. The rationale is that inhibition takes time to occur and should be more reflected in the response for slower than shorter decisions, leading to changes of compatibility effects over time following a negative slope. The same visualization was applied here for distractor affordance similarity effects. As highlighted on Fig 7, delta plots of affordance similarity effects on compatible trials also show a negative slope, reflecting increased interference from similar distractors for longer response times. Such increase over time was not observed on incompatible trials. The pattern observed in the compatible condition parallels what has been reported in the literature on inhibitory control in compatibility tasks.

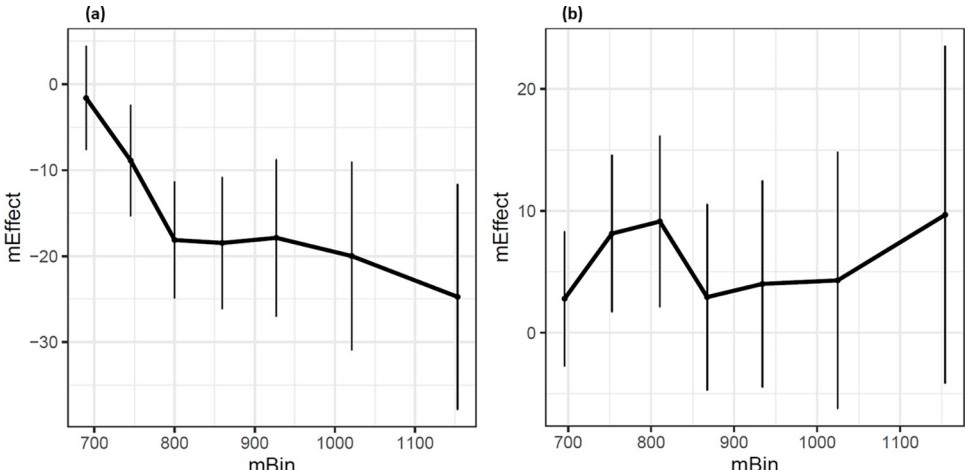

**Fig 7.** Delta plot of the effect of Similarity for (a) compatible and (b) incompatible target and response for data from Experiment 1 and 2. mEffect corresponds to the difference of response times between the dissimilar and similar conditions as a function of response time distribution (mBin). Error bars correspond to the standard errors.

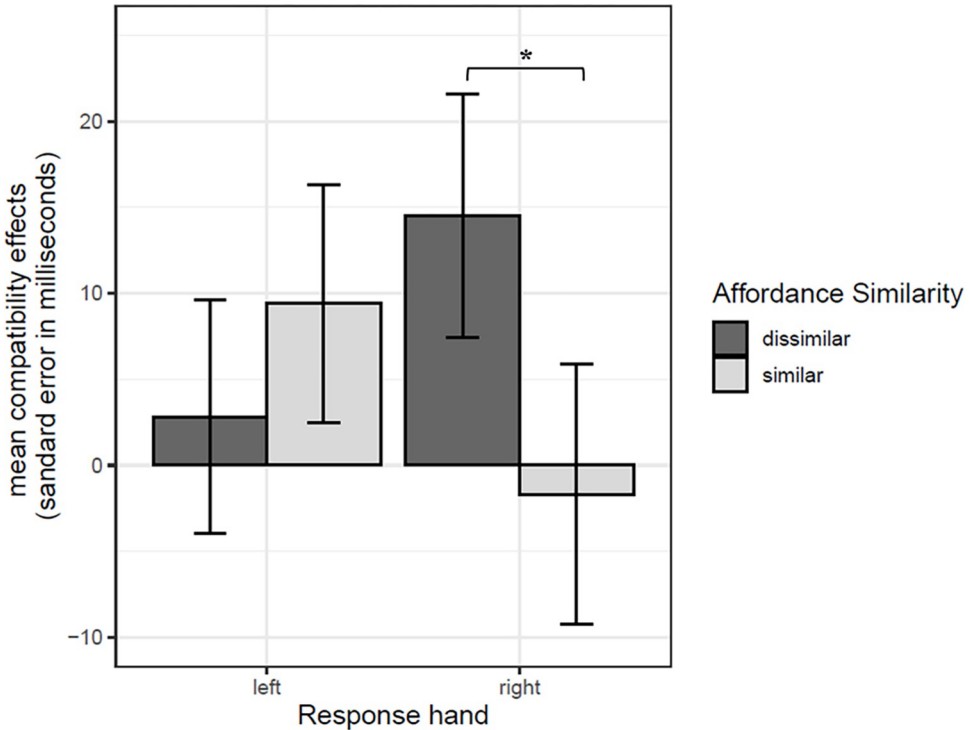

**Fig 8. Compatibility effects (mean RTs for the difference between incompatible and compatible trials) as a function of Similarity and Response Hand.** * $p < .05$. Error bars correspond to the standard errors for participants.

**Complementary analysis on compatibility effects.** An additional analysis was conducted using linear mixed models on compatibility effects as a function of Similarity, Response Hand and Experiment Version. Compatibility effects were computed as the response times difference between the incompatible and compatible conditions. The mixed model structure was chosen following the same procedure as explained before. Effect sizes were computed as Westfall's d. Fixed effects corresponded to Similarity, Response Hand and Experiment Version. The random structure included random effects for participants nested in Task version and Experiment. The model structure can be found in S2 Appendix.

The three-way interaction between Similarity, Response Hand and Experiment Version was not significant (estimate = -1.374, $t$ = -0.155, SE = 8.845, $p$ = 0.876, Westfall's d = 0.005). However, the two-way interaction between Similarity and Response Hand was significant (Fig 8; estimate = -14.622, $t$ = -2.338, SE = 6.255, $p$ = 0.019, Westfall's d = 0.051). For right hand responses, paired comparison showed greater compatibility effects when target and distractor evoked dissimilar handle affordances ($M$ = 14.51, $SD$ = 107.65), in comparison to similar affordances ($M$ = -1.70, $SD$ = 114.86; estimate = -20.850, $z$ = -2.355, SE = 8.50, $p$ = 0.0185, Westfall's d = 0.073). This effect was not found for left hand responses: compatibility effects for target and distractor evoking dissimilar handle affordances ($M$ = 2.80, $SD$ = 103.25) were not significantly different from compatibility effects for target and distractor evoking similar handle affordances ($M$ = 9.39, $SD$ = 105.08; estimate = 8.390, $z$ = -2.355, SE = 8.83, $p$ = 0.3422, Westfall's d = 0.029). Finally, no significant main effects were found for Similarity (estimate = 5.244, $t$ = 1.406, SE = 3.730, $p$ = 0.161, Westfall's d = 0.022), Response Hand (estimate = -1.018, $t$ = -0.230, SE = 4.423, $p$ = 0.817, Westfall's d = 0.005) and Experiment Version (estimate = -4.345, $t$ = -0.824, SE = 2.275, $p$ = 0.411, Westfall's d = 0.021).

## General discussion

In the present study, we addressed whether the similarity of handle affordances between objects could influence target object selection among distractors. Photographs of pairs of objects, one tool and one kitchen utensil, were presented and participants had to select the tool or the utensil by pressing keys on the side of the target object. Following the inhibition hypothesis [28, 29], we expected that participants would be faster to select the target when handle affordances of target and distractor were dissimilar, in comparison to similar, especially when target orientation and response were compatible. In the first experiment, we failed to observe an effect of distractor affordances on target processing. However, a strong general advantage of the left non-dominant response hand also corresponding to the left location of the target was found. In the second experiment, we aimed at determining whether this general left advantage could be due to a visuo-attentional bias. By introducing catch-trials requiring attention to both objects of the scenes, we expected a reduction of this putative bias. Catch-trials did not attenuate the left advantage, suggesting that the effect originated in the response selection rather than in a visuo-attentional bias. The reasons underlying overall faster motor responses with the left, non-dominant hand regardless of handle affordances of target and distractor objects remain to be elucidated. The most parsimonious interpretation relies on the bio-mechanical constraints of the hand postures involved in the key press responses, that might be not completely equivalent between left and right hands (E-C vs I-N presses respectively).

Critically, results of the two experiments combined showed an interaction between target-distractor affordance similarity, target-response compatibility and response hand. When participants had to respond with their right dominant hand and when target and response were compatible, they were slower to select the target object presented with a distractor object with similar compared to dissimilar affordances. This cost for dissimilar affordances was not found for left hand responses or when target and response were incompatible. In other words, we found that it is the similarity (and not the dissimilarity) of distractor affordances that slows down processing of a target object presented among distractors. We will come back to the theoretical implications of this main finding later in the discussion.

Interestingly, we were able to observe an effect of distractor affordances on target processing in compatible situations with key press responses. This is a major contribution of our study as it was suggested that reach-and-grasp responses were necessary to potentiate the activation of action components as they are more action relevant than key presses [9, 26]. This activation of action components may have been due to the specific key presses used in our experiment. The hand posture of the participants mimicked a grasp while pressing the keys, which might have potentiated the action more than mere key presses. In addition, the exposure to a large set of familiar graspable objects may have been sufficient to potentiate the activation of action components from visual objects, even without actual grasping responses. Indeed, previous studies investigating the influence of distractor affordances on target processing used either basic three-dimensional geometric objects (i.e., cylinders and cubes; [27]) or only very few simple objects (i.e., cups and drawer handles; [26]).

One may argue that the effect of distractors on target processing could not necessarily be due to affordance activation but could be explained in the light of the abstract coding hypothesis. This interpretation appears legitimate, especially considering the on-going debate on the nature of compatibility effects. However, considering that the abstract coding hypothesis predicts a facilitation when stimulus and response abstract properties match, opposite results would have been expected with an interference of distractors evoking dissimilar affordances on target processing. This is what was found in action-irrelevant situations in Pavese and Bux-baum's study [26] and interpreted as a saliency effect: when participants responded with key

presses, distractors were more interferent when they had dissimilar action properties to the target, compared to similar properties. In the present study, the pattern of results was in the opposite direction with an interference from distractors with similar action properties. More-over, this interference was restricted to specific conditions, which would be difficult to explain from an abstract coding perspective. Accordingly, it was restrained to responses with the dominant hand in situations of stimulus-response compatibility. This lends further support to the assumption that the response needs to be relevant to act toward the target in order to observe any effect of distractor handle affordances.

Furthermore, the fact that the impact of distractor affordances on object selection was restricted to responses made with the dominant right hand is particularly interesting. Numerous studies have highlighted differences between the dominant right hand and the left hand in action selection tasks, when objects evoke affordances, with the dominant right-hand generally more sensitive to affordances [48, 50, 51]. This greater susceptibility of the dominant hand to affordance effects might be related to the better performance of the dominant right hand in comparison to the left hand in motor coordination, motor execution and motor planning [50, 52]. Yet in the present study, the right dominant hand was more sensitive to affordances effects despite overall faster motor responses with the left hand, suggesting that the potentiation of grasp components from visual objects may be relatively independent from general response speed.

Overall, results extend previous findings reported by Ellis et al. [27] and Pavese and Bux-baum [26] on affordance and object selection to the perception of diverse complex tools and utensils. Critically, we provide additional evidence that in a context of object selection in multi-object scene perception, the pattern of compatibility effects obtained is in line with Cali-giore et al. [28] and Vainio and Ellis [29] predictions. It would imply that an inhibition mechanism is at play, with an inhibition of the target affordance when both target and distractor have similar affordances. However, while the inhibition hypothesis appears plausible in the light of our results, the present study does not allow to directly test the cognitive and brain mechanisms behind the observed behavioral cost of distractors with similar handle affordances. An alternative interpretation based on competition may explain the interference effect from distractors with similar affordances during target selection reported here. When analyzing data from the perspective of compatibility effects, we observed greater compatibility effects when target and distractor evoked dissimilar affordances in comparison to similar affordances for right hand responses. One could argue that when affordances of target and distractor are dissimilar, only the affordance of the target object is compatible with the response. In that case scenario, no competition occurs between the affordances of the target and distractor and a classical compatibility effect is observed. In contrast when target and distractor evoke similar affordances, both handle affordances compete for right hand selection, which leads to slower compatible responses and therefore a reduction of compatibility effects. Such competition would not be visible for left hand responses, as the left hand would be less sensitive to compatibility effects overall [48, 50, 51].

While both the inhibition and competition hypotheses are plausible, the evolution of the effect of similarity as a function of response time distribution (Fig 7) may offer additional elements of interpretation for the type of mechanism underlying interference form distractors with similar affordances. When target and response were compatible, the difference between similar and dissimilar affordances changed along response time distribution: the effect turned more and more negative as response times increased. The temporal dynamics of the effect of affordance similarity suggest an increase of inhibition over time, as proposed by Wildenberg et al. [49]. When target and response were incompatible, this temporal pattern was not observed, which further supports the inhibition hypothesis. Further investigation will be

needed to better characterize the mechanisms involved in the cost entailed by similar distractor affordances and its relation to inhibition processes.

This study provides novel evidence on the influence of the similarity of affordances evoked by multiple objects on object perception. Yet, it is important to emphasize some limitations to our results. First, we neither observe an overall effect of target-response compatibility nor an influence of distractor affordances on object selection independently of response hand. The impact of target-response compatibility on response times may have not been strong enough to overcome the modulations entailed by the other factors of interest, namely response hand and distractor affordances. Second, the general advantage for the left hand/hemifield remains difficult to explain, although the effect probably originates at the motor rather than visuo-attentional level. Finally, while the interaction between target-response compatibility, distractor affordance similarity and response hand was significant, the effect size for this interaction was small. Further investigations would be helpful to obtain a clearer view of the pattern of affordance effects reported in this present study, especially how the effect of multiple affordances on object selection is modulated by the response hand. Follow up studies could benefit from a greater number of trials per condition to maximize statistical power.

In conclusion, we found that in situations of multi-object scene perception, when having to select a target object among distractors objects, distractors with handle affordances similar to the target interfere with target processing and slow down target selection. This interference only appears in action-relevant situations, namely when target orientation and response are compatible, and when the response is made with the dominant hand. Results seem in line with a theoretical view proposing the existence of a mechanism of automatic inhibition of the affordance of distractors [28, 29]. Furthermore, they provide novel evidence regarding how the evocation of object affordances contributes to compatibility effects and stress the relevance of studying object affordances in complex, multi-object perceptual situations.

## Supporting information

**S1 Appendix. List of objects pairs.**
(DOCX)

**S2 Appendix. Mixed models structures and R syntaxes.**
(DOCX)

**S3 Appendix. Tables.**
(DOCX)

## Acknowledgments

The authors thank Paul Kozieja for creating the stimuli and Laurent Ott for helping with experiment programming.

## Author Contributions

**Conceptualization:** Lilas Haddad, Yannick Wamain, Solène Kalénine.

**Data curation:** Lilas Haddad.

**Formal analysis:** Lilas Haddad, Solène Kalénine.

**Methodology:** Lilas Haddad, Yannick Wamain, Solène Kalénine.

**Resources:** Lilas Haddad, Yannick Wamain, Solène Kalénine.

**Supervision:** Yannick Wamain, Solène Kalénine.

**Visualization:** Lilas Haddad.

**Writing – original draft:** Lilas Haddad.

**Writing – review & editing:** Lilas Haddad, Yannick Wamain, Solène Kalénine.

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
