## [Decision Letter · Decision Letter 0]

20 Apr 2023

PONE-D-23-04726Too much to handle? Interference of distractors with similar affordances on target selection for handled objectsPLOS ONE

Dear Dr. Kalénine,

Thank you for submitting your manuscript to PLOS ONE. After careful consideration, we feel that it has merit but does not fully meet PLOS ONE’s publication criteria as it currently stands. Therefore, we invite you to submit a revised version of the manuscript that addresses the points raised during the review process. First of all I apologize for the delay of the review process. The paper has been evaluated by 3 experts in the field, and all found considerable merit to your manuscript. However, all reviewers reported some issues in the methodology of data collection, statistical analyis, model that is proposed for the interpretation of the results and framing in the context of current knowledge. Such issues need to be addressed before the manuscript can be accepted in its rightful postion in the literature.

We look forward to receiving your revised manuscript.

Kind regards,

Luigi Cattaneo, MD, PhD

Academic Editor

PLOS ONE

2. Please provide additional details regarding participant consent. In the ethics statement in the Methods and online submission information, please ensure that you have specified (1) whether consent was informed and (2) what type you obtained (for instance, written or verbal, and if verbal, how it was documented and witnessed). If your study included minors, state whether you obtained consent from parents or guardians. If the need for consent was waived by the ethics committee, please include this information. If you are reporting a retrospective study of medical records or archived samples, please ensure that you have discussed whether all data were fully anonymized before you accessed them and/or whether the IRB or ethics committee waived the requirement for informed consent. If patients provided informed written consent to have data from their medical records used in research, please include this information.

Reviewers' comments:

Reviewer's Responses to Questions

**Comments to the Author**

1. Is the manuscript technically sound, and do the data support the conclusions?

Reviewer #1: Partly

Reviewer #2: Yes

Reviewer #3: Partly

2. Has the statistical analysis been performed appropriately and rigorously? 

Reviewer #1: Yes

Reviewer #2: Yes

Reviewer #3: Yes

3. Have the authors made all data underlying the findings in their manuscript fully available?

Reviewer #1: Yes

Reviewer #2: Yes

Reviewer #3: Yes

4. Is the manuscript presented in an intelligible fashion and written in standard English?

Reviewer #1: No

Reviewer #2: Yes

Reviewer #3: Yes

5. Review Comments to the Author

Reviewer #1: The paper reports the results of two online experiments where the authors investigate the impact of distractor affordances on the identification of a target object. In each trial two familiar graspable objects were presented. Both objects had a handle that could be oriented to the left or right, congruently or incongruently, resulting in similar or dissimilar handle affordances. Participants had to determine on which side the target was presented by simultaneously pressing two keys located on the same side. The idea behind this response pattern was that the participants’ hand posture would mimic a grasp. The two experiments were quite similar, except that catch trials were also introduced in Experiment 2. Results from the two combined experiments show that affordance similarity (rather than dissimilarity) between target and distractor slows down dominant hand responses, in line with an interference from distractors with similar action properties.

The present study has the merit of analyzing aspects still relatively understudied in the literature regarding the effect of distractor affordance on target processing. The use of a large set of familiar objects and a response mode in which the hand posture would mimic a grasp are methodological improvements compared to previous studies.

However, it presents critical issues that require further efforts for its publication.

Introduction

The introduction is very long and sometimes difficult to follow. It could benefit from a more focused approach to the research question. For example, the paragraph ranging from line 85 to 105 could be shortened since it addresses a topic that is out of the issue posed by the authors. Furthermore, an important piece of literature that investigated the relationships between multiple objects in the visual scene (The paired-object affordance effect, Yoon et al., 2010; see also Federico & Brandimonte, 2019; Borghi et al., 2012) is completely absent and should be presented to the readers and also discussed. The literature about action inhibition related to non-target objects should also be implemented (see for example Vainio et al., 2022; 2014; Vainio, 2021; see also Garofalo & Riggio, 2022).

Method

Power analysis

The power analysis paragraph lacks some important indexes, such as the level of alpha and beta and which function of pwr was used.

Analyses and Results

I overall appreciate the use of linear mixed modelling approach. However, there are some aspects of the analysis that are not clear to me. In detail, why the random structure of the models (both for accuracy and reaction times) is different between the experiments?

Looking at the results, despite the absence of the critical interaction in the single experiments, it appears clear to me that the orientation of the objects differentially affected the hand response. In detail, I calculated the compatibility effect in both experiments (see below).

right hand left hand

Similar Dissimilar Similar Dissimilar

Exp1 Compatibility effect

(Incompatible - Compatibile) -5 16 6 7

Exp2 Compatibility effect

(Incompatible - Compatibile) -5 16 3 2

An alternative interpretation of this data pattern is as follows: The right hand is more sensitive to handles, whether they occur in the same (similar) or opposite (dissimilar) orientation. When both handles have the same orientation, there is a competition between them leading to slower compatible responses and to negative handle-hand compatibility effects, while when the orientation is dissimilar, only the handle of the object compatible with the right hand affects performance, leading to a standard compatibility effect. In contrast, the left hand shows an overall and similar reduced effect for the two object orientations, and when the task load is further increased, as in Experiment 2, there is a further reduction in the residual effect.

I would suggest authors to consider this alternative explanation and consider performing the analysis also on the compatibility effects.

Furthermore, many subjects are used for each experiment, but despite this, the significance of the interaction of interest is achieved only by combining the data of the two experiments. This probably derives from the online procedure used in the experiments which does not allow a control of the conditions in which the participant performs the task. Indeed, the data show considerable inter-subject variability. With regard to this, one wonders how far the results are comparable with the studies cited on the topic. Perhaps it would be appropriate to repeat the experiment with a standard laboratory procedure, reducing the number of participants and increasing the number of trials in order to reduce the inter-subjective variability.

Discussion

The main result of the study concerns the cost associated with the right hand in compatible trials when target and distractor have similar rather than dissimilar affordances. This cost is not found for left hand responses or when target and response are incompatible. This result is probably a further example of the fact that the right/dominant hand has an advantage in interfacing with objects in many aspects of motor behavior. The difference between the hands, reported in the paper, is not really discussed and explored also in comparison to the various studies on this topic (see for example Riddoch et al. 1998; Fischer & Dahal, 2007; Hughes et al., 2011). I think the paper could benefit from this aspect of the discussion.

Finally, I recommend authors to re-read the text carefully because there are many typos and to check the citations and references because there are deficiencies and inconsistencies.

References

Borghi, A. M., Flumini, A., Natraj, N., & Wheaton, L. A. (2012). One hand, two objects: Emergence of affordance in contexts. Brain and cognition, 80(1), 64-73.

Federico, G., & Brandimonte, M. A. (2019). Tool and object affordances: An ecological eye-tracking study. Brain and cognition, 135, 103582.

Fischer M. H., & Dahl C. D. (2007). The time course of visuo-motor affordances. Experimental Brain Research, 176, 519–524.

Hughes C. M. L., Reißig P., & Seegelke C. (2011). Motor planning and execution in left- and right-handed individuals during a bimanual grasping and placing task. Acta Psychologica, 138, 111–118

Garofalo, G., & Riggio, L. (2022). Influence of colour on object motor representation. Neuropsychologia, 164, 108103.

Riddoch J. M., Edwards M. G., Humphreys G. W., West R., & Heafield T. (1998). Visual affordances direct action: Neuropsychological evidence from manual interference. Cognitive Neuropsychology, 15, 645–683

Yoon, E. Y., Humphreys, G. W., & Riddoch, M. J. (2010). The paired-object affordance effect. Journal of Experimental Psychology: Human Perception and Performance, 36(4), 812.

Vainio, L., Ala-Salomäki, H., Huovilainen, T., Nikkinen, H., Salo, M., Väliaho, J., & Paavilainen, P. (2014). Mug handle affordance and automatic response inhibition: Behavioural and electrophysiological evidence. Quarterly Journal of Experimental Psychology, 67(9), 1697-1719.

Vainio, L. (2021). Automatic inhibition of habitual response associated with a non-target object while performing goal-directed actions. Quarterly Journal of Experimental Psychology, 74(4), 716-732.

Vainio, L., Tiippana, K., Peromaa, T., Kuuramo, C., & Kurki, I. (2022). Negative affordance effect: automatic response inhibition triggered by handle orientation of non-target object. Psychological Research, 1-14.

Reviewer #2: This paper reports 2 experiments which show left-visual field advantage for responses, but results did not support the authors’ hypothesis that responses would be faster when target and distractor afforded opposite dissimilar actions relative to the same action. A second experiment ruled out a possible explanation for the left-visual-field advantage being due to participant strategy/attentional bias.

The paper and data are clearly presented, and goes some way to answer questions about inhibition of unwanted actions afforded by objects. Indeed, I have wondered how competition between actions evoked by multiple affordances might be resolved, as it seems that objects with affordances incompatible with the required response are evoked automatically (e.g., https://journals.sagepub.com/doi/10.1080/17470218.2011.588336).

The paper, methods, and analysis appear solid, and the findings make a contribution to the field.

Recommended before publication

Authors report that they removed RT outliers (pg 16, 370 onwards) according to particular criteria. This is common when working with R data because they are typically non-normally distributed and outliers skew analyses. The criteria chosen seem reasonable on the face of it, but I wonder if there was any particular rationale for the particular cut-offs that were chosen? If there was not an a-priori reason to choose these, as the precise method chosen can bias findings (see e.g., https://www.frontiersin.org/articles/10.3389/fpsyg.2021.675558/full) I’d be reassured that the findings reported are not a serendipitous artefact of this method if the authors could repeat their analysis – using a different method to deal with the well-known non-normality of RT data – and report qualitatively the same result (perhaps include as a footnote in the manuscript?). For different approaches, see Ratcliff, R., 1993. Methods for dealing with reaction time outliers https://psycnet.apa.org/doiLanding?doi=10.1037%2F0033-2909.114.3.510

Recommended:

The authors mention the possibility of inhibition of an afforded response when target and distractor are associated with the same response. If so, I would expect that inhibition to take time to develop and so we might see evidence of this by examining compatibility effects across the RT distribution. We might expect positive effects which gradually turn negative at later portions of the RT distribution if the data were plotted as a delta plot – as commonly shown in other “conflict” tasks (see van den Wildenberg et al., 2010 for a nice review of the technique and its advantages; https://www.frontiersin.org/articles/10.3389/fnhum.2010.00222/full).

I’m not necessarily surprised that correct responses were not overall faster when distractors afforded an action that was incompatible, when target and response were compatible. This reminds me of Eriksen Flanker tasks (e.g., https://link.springer.com/article/10.3758/BF03203267, as one example but there are many others) – where participants are typically faster to respond when target and flakers (distractors) are associated with the same response (congruent) relative to different responses (incongruent). Perhaps there is a similar mechanism, here, whereby distractors affording responses opposite the target create competition which needs to be resolved? It might be helpful for the authors to make theoretical links to other conflict tasks, or to explain why they believe their task/mechanism to be different here.

Reviewer #3: line 84: I believe this matter is still very debated, so such a conclusion is not warranted

line 249: they were informed about the aim of the study: what kind of details were provided?

line 277: total stimuli 96 stimuli / 8 configurations = 12 stimuli per configuration. This might partially be the reason the lack of effects in the the two (separate) experiments….is there a reason behind this choice? 12 stimuli per “design cell” are very few.

Why are the trials so short? the transition between the 3 scenarios is extreme.

exclusion of responses >4000 ms…It looks like even 2500 ms is enough to provide a response…why 4000ms?

The power analysis reported indicates that 120 participants were necessary to get an effect. However, participants resulting form exp 1 and 2 are ~60. Moreover, the authors mentioned the power but not the alpha level (<0.05?) and the tails of the hypothesis (which I assume is 2).

Instead of changing the data with the 2SD rule (which is not optimal for outlier detection), the authors might use the medians instead of the means so that big outliers should not affect the central tendency measure. Considering the small number of trials per cell I think this is the more elegant solution. Otherwise check Jones 2019 https://link.springer.com/article/10.3758/s13414-019-01726-3 for other strategies.

The authors found an effect, that partially fit their hypothesis. This effect however arises from the total sample of participants who participated in both experiments. I would like the authors to calculate and report the effect size of their results

Curiously the authors did not find an affordance effect independently from the distractors (i.e. produced by the handle orientation itself, the compatibility). Did the authors expected that?

The results part is not written properly in my opinion. It would be much better if the main effects are described before the interactions, and the interactions with a lower number of factor would be described before the ones with a higher number of factors (two-way before three way).

It would also be advisable to properly and explicitly describe the design (es 2x2x2 with three within independent variables, including factor1(level a, level b), factor2 (level a level b) etc… ).

I think that in general the manuscript could be streamlined: es.: line 318 some of the description has already been mentioned before

In the discussion little or no space has been given to the unexpected result of related to the faster left hand responses compared to right hand ones.

line 592: there is an asterisk which should not be there I believe :-)

line 614: response selection (instead of response hand)

line 618: “identify” might be misleading: one might intepret it as the motor representations facilitated by the affordance effect are necessary to identify the object as a kitchen tool or an utensil. This is not necessarily the case. I would use “select” instead (also in other sections of the manuscript)

line 297 and 628: one can turn the argument the other way around: this strategy might have been detrimental in letting the effects emerge: the shape of the responding fingers might be quite important, but the one depicted in the figure is quite far from the final position that could be adopted by a person grasping one of these objects. It resambles more a precision grip with two fingers. The visual resamblance between the participants’ fingers shape and the one that would have used in order to grasp the object does not necessarily help the facilitation of the motor representation of a whole hand grasp, on the contrary, it might have inhibited the “object-grasp hand shape”, because a pinch-like gasping fits different type of objects (seldom one would grasp a whisker with hands shaped like those in figure 2). In general, when looking for automatic visuomotor effects, the best motor responses to be selected for the task are those that fit/resamble the most the object/action observed (a general argument for stimulus response compatibility effects: Kronblum 1990 10.1037/0033-295X.97.2.253; Barchiesi & Cattaneo 2015 10.1016/j.neuropsychologia.2015.01.030 on automatic visuomotor effects).

line 641: The task lacks a control for the affordance effect. For example, if the effect is genuine (and not spatially driven by the handle directions for example), then, the authors should have expected a modulation of the effects if the shape of the fingers was changed, or if abstract oriented stimuli were added as control condition.

The authors claim that the “interference by similarity effect” found combining the two experiments rules out the “abstract spatial effects” explanation even though the objects are oriented shapes. According to the authors this claim is strengthened by the result obtained, i.e., that the interference effect is found only for the right hand. Again the argument in favour of the results can be turned against: the authors did not predict a right hand effect specifically for the compatibility x distractors effect. The interpretation of the three-way interaction is a post-hoc one. So, who knows whether using other abstract stimuli as a control condition the authors might have got similar effects as those obtained with the objects…

line 650: ok, but if this is true, then how do the authors explain the left hand speed? I would expect that right hand ideal conditions (dissimilar distractor compatible handle) would produce the fastest reaction times within the experiment…which is not the case…unless

line 658: it is not necessarily an inhibition mechanism…it could be a competition mechanism right?

When in time do the authors expect an affordance effect? the average responses were very delayed (~800 ms). I was wondering whether the task was so difficult that the participants responses were not very influenced anymore by the affordances. On other visuomotor effects these are very early in time (Barchiesi & Cattaneo 2015 10.1016/j.neuropsychologia.2015.01.030).

I would like the authors to add a limitation paragraph including:

the partial confirmation of their hypothesis (no compatibility main effect, no interaction between compatibility and distractors)

left hand responses are faster in general (and add speculations about it)

the (most likely) small effect size of their results in terms of distractor similarity

how the small number of trials per cell could have been accounted for the lack of fully expected effects

6. PLOS authors have the option to publish the peer review history of their article (what does this mean?). If published, this will include your full peer review and any attached files.

Reviewer #1: No

Reviewer #2: No

Reviewer #3: No

---

## [Author Response · Author response to Decision Letter 0]

26 Jun 2023

Response letter to the comments of the reviewers

Reviewer: 1

The paper reports the results of two online experiments where the authors investigate the impact of distractor affordances on the identification of a target object. In each trial two familiar graspable objects were presented. Both objects had a handle that could be oriented to the left or right, congruently or incongruently, resulting in similar or dissimilar handle affordances. Participants had to determine on which side the target was presented by simultaneously pressing two keys located on the same side. The idea behind this response pattern was that the participants’ hand posture would mimic a grasp. The two experiments were quite similar, except that catch trials were also introduced in Experiment 2. Results from the two combined experiments show that affordance similarity (rather than dissimilarity) between target and distractor slows down dominant hand responses, in line with an interference from distractors with similar action properties. The present study has the merit of analyzing aspects still relatively understudied in the literature regarding the effect of distractor affordance on target processing. The use of a large set of familiar objects and a response mode in which the hand posture would mimic a grasp are methodological improvements compared to previous studies. However, it presents critical issues that require further efforts for its publication.

1) The introduction is very long and sometimes difficult to follow. It could benefit from a more focused approach to the research question. For example, the paragraph ranging from line 85 to 105 could be shortened since it addresses a topic that is out of the issue posed by the authors. 

Response: We thank the Reviewer for the feedback on our introduction section. As suggested, we have shortened some of the arguments (page 4, line 88): “However, recent findings indicated that the combination of specific types of task (e.g., a task relevant for action) and response (e.g., reach-and-grasp response) could favour the activation of action components (9). The affordance hypothesis is also legitimated by neurophysiological studies highlighting an activity of the motor system during the perception of manipulable objects, independently of the SRC paradigm (10–13).”

In addition, we shortened several paragraphs of the Introduction, namely the section about:

-competition between multiple affordances evoked by a single object (page 5, line 101).

-empirical data on the evocation of multiple affordances evoked by different objects (page 8, line 175).

2) Furthermore, an important piece of literature that investigated the relationships between multiple objects in the visual scene (The paired-object affordance effect, Yoon et al., 2010; see also Federico & Brandimonte, 2019; Borghi et al., 2012) is completely absent and should be presented to the readers and also discussed. The literature about action inhibition related to non-target objects should also be implemented (see for example Vainio et al., 2022; 2014; Vainio, 2021; see also Garofalo & Riggio, 2022).

Response: Since we did not manipulate the use relationships between objects in the present study, we initially did not review the paired-object affordance effects, but we agree that this literature is relevant regarding affordance perception in multi-object situations in general. It has been added on page 6, line 123: “Previous research on affordance activation in multi-object perceptual situations is very limited. In multi-object situations, the other objects in the scene provide a context and potentiate the way we perceive a given object (21–23). For instance, when an object-tool pair is presented within a visual scene, each object in the pair does not only activate the action possibilities it would typically afford when presented alone but also those associated with the common or uncommon use of the tool in conjunction with the specific object from the pair. This is the case when a knife is situated near a screw, it may suggest the action of "screwing" rather than the typical action of "cutting." However, it is far from clear whether competition phenomena arise from distinct affordances evoked by multiple objects”

Regarding the literature on action inhibition of non-target objects, we have added empirical evidence to the action inhibition proposal presented on pages 9 line 197: “Predictions of the inhibition hypothesis have been supported by some empirical data (30–33). In one study, Vainio et al. (31) presented to participants an object with its handle oriented for a right or left-hand grasp. The object served as a prime of a target line or target arrow oriented to the left or to the right. In a go-no go task, participants had to refrain to answer when the target was a line but had to determine the direction of the arrow by pressing with their left thumb if the arrow pointed to the left and with their right thumb if the arrow pointed to the right. Overall, results showed that participants took longer to judge the direction of the target arrow when it was presented in an orientation similar to the handle of the non-target object prime, as compared to dissimilar. In addition, in no-go trials, participants tended to incorrectly respond to the target more when target and prime objects were dissimilarly oriented, as compared to similarly oriented. These results are in line with the predictions of the inhibition hypothesis, as distractor objects with orientation properties similar to the target and response seem to interfere with target processing, more than distractor evoking dissimilar properties. Furthermore, errors in no-go trials provided additional support in favor of a mechanism based on affordance inhibition: participants had more difficulty to refrain from responding when distractor affordances were dissimilar. However, although the few empirical data presented are consistent with inhibition hypothesis, the different predictions still need to be investigated with scenes of familiar objects.”

3) The power analysis paragraph lacks some important indexes, such as the level of alpha and beta and which function of pwr was used.

Response: We apologize for this omission. We used the function pwr.t.test of the pwr package. Regarding the different indexes of the power analysis, we considered a beta level of 0.2 and therefore a power (1-beta) of 0.8 for an alpha level of 0.05. 

We have specified this information on page 11 line 259 and page 12, line 263:

“An a priori power analysis was conducted with R software using the function pwr.t.test of the pwr package (v1.3-0; (35)).”

“To guarantee a sufficient statistical power for a two-tails hypothesis (β = 0.2; power (1- β) = 0.80; α = 0.05), about 120 participants were anticipated.”

4) I overall appreciate the use of linear mixed modelling approach. However, there are some aspects of the analysis that are not clear to me. In detail, why the random structure of the models (both for accuracy and reaction times) is different between the experiments?

Response: We admit that the data analysis section is complex and that we may have not provide enough explanation regarding selection of random structures in mixed-effect. The random structures of the models are different between experiments because we chose the maximum random structure supported by the data, following the guidelines proposed by Barr et al. (2013) and Bates et al. (2015). We started from the maximal random structure and if the models did not converge, we reduced the random structures. To reduce it, we ran Principal Component Analyses to estimate the part of variance of our model explained by each intercept and slope with the rePCA function from the lme4 package (v1.1-27.1; Bates et al., 2015). We only kept the intercepts and slopes explaining the biggest part of variance, repeating this process until the models converged. As data are different between experiments, the models fitting the different data sets the best may therefore have different random effect structures. 

We have added details about the random- effect structures of mixed models on page 17, line 393: “To choose random structures, we followed the guidelines proposed by Barr et al. and Bates et al. (44,45). We first built our model with the maximal random structure possible. If the model did not converge, we reduced the random structure. To do so, we ran Principal Component Analyses to estimate for each intercept and slope of our model the part of variance explained. We used the rePCA function from the lme4 package (v1.1-27.1; (39)). We kept the intercepts and slopes that explained the biggest part of variance and removed the ones explaining only a small percentage of variance. This process was repeated until the model converged.”

5) Looking at the results, despite the absence of the critical interaction in the single experiments, it appears clear to me that the orientation of the objects differentially affected the hand response. In detail, I calculated the compatibility effect in both experiments (see below).

right hand left hand

Similar Dissimilar Similar Dissimilar

Exp1 Compatibility effect

(Incompatible - Compatibile) -5 16 6 7

Exp2 Compatibility effect

(Incompatible - Compatibile) -5 16 3 2

An alternative interpretation of this data pattern is as follows: The right hand is more sensitive to handles, whether they occur in the same (similar) or opposite (dissimilar) orientation. When both handles have the same orientation, there is a competition between them leading to slower compatible responses and to negative handle-hand compatibility effects, while when the orientation is dissimilar, only the handle of the object compatible with the right hand affects performance, leading to a standard compatibility effect. In contrast, the left hand shows an overall and similar reduced effect for the two object orientations, and when the task load is further increased, as in Experiment 2, there is a further reduction in the residual effect. I would suggest authors to consider this alternative explanation and consider performing the analysis also on the compatibility effects.

Response: We have conducted analyses on compatibility effects (corresponding to the difference of RTs between incompatible and compatible conditions) as a function of affordance similarity and response hand on the data of Experiment 1 and 2 (final analysis). We report the statistics and graphical representation of the effects found below. A significant interaction was found between the similarity of affordances and response hand (estimate = -14.622, t = -2.338, SE = 6.255, p = 0.019). Paired comparisons highlighted that when participants responded to the task with their right hand, compatibility effects were greater when affordances were similar (M = 14.51, SD = 107.65), in comparison to dissimilar (M = -1.70, SD = 114.86; estimate = -20.850, z = -2.355, SE = 8.50, p = 0.0185, Westfall’s d = -0.073). No further significant effects were found (in particular, no interaction with Experiment).

For right hand responses, we found greater compatibility effects for target and distractor evoking dissimilar than similar affordances, which seem in line with an alternative competition mechanism, as proposed by the Reviewer. Indeed, when affordances of target and distractor are dissimilar, only the affordance of the target object is compatible with the response. In that case scenario, no competition occurs between the two affordances and a classical compatibility effect is observed. When target and distractor evoke dissimilar affordances, both affordances compete with one another, which leads to slower compatible responses and therefore a reduction of compatibility effects. However, we do not observe a complete negative compatibility effect here, rather an absence of compatibility effect. 

Regarding left hand responses, we found no differences of compatibility effect amplitude between similar and dissimilar affordances. Following our previous interpretation of this absence of effect for left hand responses, it is possible that affordances are less salient when directed toward a left-hand response and therefore it would not be enough for a competition to arise between the two affordances of target and distractor objects.

We added the analysis and figure on compatibility effects for data of Experiment 1 and 2 combined in a complementary analysis subsection of the Results section, page 28 line 638.

We also discussed the results of this analysis page 32, line 744 to discuss this specific competition interpretation of our effect: “An alternative interpretation based on competition may explain the interference effect from distractors with similar affordances during target selection reported here. When analyzing data from the perspective of compatibility effects, we observed greater compatibility effects when target and distractor evoked dissimilar affordances in comparison to similar affordances for right hand responses. One could argue that when affordances of target and distractor are dissimilar, only the affordance of the target object is compatible with the response. In that case scenario, no competition occurs between the affordances of the target and distractor and a classical compatibility effect is observed. In contrast when target and distractor evoke similar affordances, both handle affordances compete for right hand selection, which leads to slower compatible responses and therefore a reduction of compatibility effects. Such competition would not be visible for left hand responses, as the left hand would be less sensitive to compatibility effects overall (49,51,52).”

References

Fischer M. H., & Dahl C. D. (2007). The time course of visuo-motor affordances. Experimental Brain Research, 176, 519–524.

Netelenbos, N., & Gonzalez, C. L. (2015). Is that graspable? Let your right hand be the judge. Brain and Cognition, 93, 18-25.

Riddoch J. M., Edwards M. G., Humphreys G. W., West R., & Heafield T. (1998). Visual affordances direct action: Neuropsychological evidence from manual interference. Cognitive Neuropsychology, 15, 645–683

6) Furthermore, many subjects are used for each experiment, but despite this, the significance of the interaction of interest is achieved only by combining the data of the two experiments. This probably derives from the online procedure used in the experiments which does not allow a control of the conditions in which the participant performs the task. Indeed, the data show considerable inter-subject variability. With regard to this, one wonders how far the results are comparable with the studies cited on the topic. Perhaps it would be appropriate to repeat the experiment with a standard laboratory procedure, reducing the number of participants and increasing the number of trials in order to reduce the inter-subjective variability.

Response: We agree with the Reviewer that online studies cannot directly control how participants do the task. We instructed them to stay seated in front of their computer for the entire duration of the experiment but we cannot ensure that they did the task as instructed. However, we were able to check that participants did position their hand and performed pseudo-grasp responses on their keyboard as requested (see the section about response modalities in the manuscript on page 13). Moreover, we also admit that the inter-subject variability is greater for online studies than on-site studies (which can occur also because the participants recruited online are usually more diverse than the one recruited on-site; Anwyl-Irvine et al., 2021). However, numerous studies showed that data and results of online perceptual and sensori-motor studies are actually comparable to on-site studies (Germine et al., 2012; Tsay et al., 2021; Woods et al., 2015). In addition, we choose the Pavlovia platform because it allowed us to restrict on which device the experiment could be run (in our case only on a computer). For experiments ran on computers, Pavlovia platform offers very good performances in terms of visual delay and reaction time accuracy (Anwyl-Irvine et al., 2021). Finally, we piloted Experiment 2 on 16 participants on-site. Response time distribution for this pilot (below) is very similar to the response times distribution for the online study (see comment 1 of Reviewer 2 for further details of response times distribution). 

Considering all these parameters and also the number of participants we recruited, we are confident in the results of the online study. 

References:

Anwyl-Irvine, A., Dalmaijer, E. S., Hodges, N., & Evershed, J. K. (2021). Realistic precision and accuracy of online experiment platforms, web browsers, and devices. Behavior research methods, 53, 1407-1425.

Germine, L., Nakayama, K., Duchaine, B. C., Chabris, C. F., Chatterjee, G., & Wilmer, J. B. (2012). Is the Web as good as the lab? Comparable performance from Web and lab in cognitive/perceptual experiments. Psychonomic bulletin & review, 19, 847-857.

Tsay, J. S., Lee, A., Ivry, R. B., & Avraham, G. (2021). Moving outside the lab: The viability of conducting sensorimotor learning studies online. arXiv preprint arXiv:2107.13408.

Woods, A. T., Velasco, C., Levitan, C. A., Wan, X., & Spence, C. (2015). Conducting perception research over the internet: a tutorial review. PeerJ, 3, e1058.

7) The main result of the study concerns the cost associated with the right hand in compatible trials when target and distractor have similar rather than dissimilar affordances. This cost is not found for left hand responses or when target and response are incompatible. This result is probably a further example of the fact that the right/dominant hand has an advantage in interfacing with objects in many aspects of motor behavior. The difference between the hands, reported in the paper, is not really discussed and explored also in comparison to the various studies on this topic (see for example Riddoch et al. 1998; Fischer & Dahal, 2007; Hughes et al., 2011). I think the paper could benefit from this aspect of the discussion.

Response: We agree that the manuscript was lacking this very important discussion. We added a paragraph in the Discussion section of the manuscript page 31, line 724: “Furthermore, the fact that the impact of distractor affordances on object selection was restricted to responses made with the dominant right hand is particularly interesting. Numerous studies have highlighted differences between the dominant right hand and the left hand in action selection tasks, when objects evoke affordances, with the dominant right-hand generally more sensitive to affordances (49,51,52). This greater susceptibility of the dominant hand to affordance effects might be related to the better performance of the dominant right hand in comparison to the left hand in motor coordination, motor execution and motor planning (51,53). Yet in the present study, the right dominant hand was more sensitive to affordances effects despite overall faster motor responses with the left hand, suggesting that the potentiation of grasp components from visual objects may be relatively independent from general response speed.”

8) Finally, I recommend authors to re-read the text carefully because there are many typos and to check the citations and references because there are deficiencies and inconsistencies.

Response: The new version of the manuscript has been carefully checked.

References

Borghi, A. M., Flumini, A., Natraj, N., & Wheaton, L. A. (2012). One hand, two objects: Emergence of affordance in contexts. Brain and cognition, 80(1), 64-73.

Federico, G., & Brandimonte, M. A. (2019). Tool and object affordances: An ecological eye-tracking study. Brain and cognition, 135, 103582.

Fischer M. H., & Dahl C. D. (2007). The time course of visuo-motor affordances. Experimental Brain Research, 176, 519–524.

Hughes C. M. L., Reißig P., & Seegelke C. (2011). Motor planning and execution in left- and right-handed individuals during a bimanual grasping and placing task. Acta Psychologica, 138, 111–118

Garofalo, G., & Riggio, L. (2022). Influence of colour on object motor representation. Neuropsychologia, 164, 108103.

Riddoch J. M., Edwards M. G., Humphreys G. W., West R., & Heafield T. (1998). Visual affordances direct action: Neuropsychological evidence from manual interference. Cognitive Neuropsychology, 15, 645–683

Yoon, E. Y., Humphreys, G. W., & Riddoch, M. J. (2010). The paired-object affordance effect. Journal of Experimental Psychology: Human Perception and Performance, 36(4), 812.

Vainio, L., Ala-Salomäki, H., Huovilainen, T., Nikkinen, H., Salo, M., Väliaho, J., & Paavilainen, P. (2014). Mug handle affordance and automatic response inhibition: Behavioural and electrophysiological evidence. Quarterly Journal of Experimental Psychology, 67(9), 1697-1719.

Vainio, L. (2021). Automatic inhibition of habitual response associated with a non-target object while performing goal-directed actions. Quarterly Journal of Experimental Psychology, 74(4), 716-732.

Vainio, L., Tiippana, K., Peromaa, T., Kuuramo, C., & Kurki, I. (2022). Negative affordance effect: automatic response inhibition triggered by handle orientation of non-target object. Psychological Research, 1-14.

Reviewer: 2

This paper reports 2 experiments which show left-visual field advantage for responses, but results did not support the authors’ hypothesis that responses would be faster when target and distractor afforded opposite dissimilar actions relative to the same action. A second experiment ruled out a possible explanation for the left-visual-field advantage being due to participant strategy/attentional bias. The paper and data are clearly presented, and goes some way to answer questions about inhibition of unwanted actions afforded by objects. Indeed, I have wondered how competition between actions evoked by multiple affordances might be resolved, as it seems that objects with affordances incompatible with the required response are evoked automatically (e.g., https://journals.sagepub.com/doi/10.1080/17470218.2011.588336).

1) Authors report that they removed RT outliers (pg 16, 370 onwards) according to particular criteria. This is common when working with R data because they are typically non-normally distributed and outliers skew analyses. The criteria chosen seem reasonable on the face of it, but I wonder if there was any particular rationale for the particular cut-offs that were chosen? If there was not an a-priori reason to choose these, as the precise method chosen can bias findings (see e.g., https://www.frontiersin.org/articles/10.3389/fpsyg.2021.675558/full) I’d be reassured that the findings reported are not a serendipitous artefact of this method if the authors could repeat their analysis – using a different method to deal with the well-known non-normality of RT data – and report qualitatively the same result (perhaps include as a footnote in the manuscript?). For different approaches, see Ratcliff, R., 1993. Methods for dealing with reaction time outliers https://psycnet.apa.org/doiLanding?doi=10.1037%2F0033-2909.114.3.510

Response: We agree that choosing how to trim response times is not trivial and may influence the results observed. Dealing with non-normality of RT distribution in mixed models is particularly challenging, as certain relevant functions to model RT distribution are not implemented in classical mixed-model packages. For example, we considered applying an ex gaussian (instead of gaussian) function to our response times, but it was not possible to use it in generalized mixed-effect models (glmer) in R. We thought about not trimming the data at all or to even apply a log transformation to the data in order to reduce the skewness of RTs distribution, but we found that applying a mean/SD procedure on raw RTs was the best compromise to reduce the skewness of the distribution, keep a maximum of data, and keep the model estimates easily interpretable. You may find below the three RT distribution with our current trimming method, log transformation on trimmed data, no trimming and trimmed data using the Median Absolute Deviation (MAD) method (as suggested by Reviewer 3) for both Experiments:

Experiment 1:

Experiment 2:

Critically, the trimming method did not impact our results. When the same mixed models are conducted on mean/SD trimmed data (the trimming method chosen), log transformed trimmed data and MAD trimmed data, the same result pattern is observed. 

We report below the results for the three way-interaction between Similarity, Compatibility and Response hand for combined experiments 1 and 2:

Trimmed RTs: estimate = -9.877, t = -2.279, SE = 4.334, p = 0.022

Log trimmed RTs: estimate = -0.009, t = -2.337, SE = 0.004, p = 0.019

MAD trimmed RTs: estimate = -9.184, t = -2.449, SE = 7.750, p = 0.014

Overall, results are consistent with a selective effect of affordance similarity regardless of the trimming method and data transformation. Regarding the skewness of the distribution, trimming the data seems better than not trimming. Applying a log transformation does also reduce skewness in comparison to non-transformed trimmed data. However, it is difficult to interpret estimates with log-transformed data. Finally trimming according to MAD removed a larger proportion of the data (see our response to Reviewer 3). Therefore, we preferred to keep untransformed RTs and a trimming procedure based on mean/SD. We nonetheless mention in a footnote page 17 that the result pattern was similar with the alternative trimming procedures: “For all experiments, the same patterns of results were observed with alternative RT pre-processing procedures such as trimming based on Median Absolute Deviation (MAD) or analyses on log-transformed data. Analyses on non-transformed RTs after trimming based on mean/SD was chosen as the best compromise to simultaneously consider the skewness of the RT distribution, the interpretation of the model estimates and the proportion of excluded trials.”

2) The authors mention the possibility of inhibition of an afforded response when target and distractor are associated with the same response. If so, I would expect that inhibition to take time to develop and so we might see evidence of this by examining compatibility effects across the RT distribution. We might expect positive effects which gradually turn negative at later portions of the RT distribution if the data were plotted as a delta plot – as commonly shown in other “conflict” tasks (see van den Wildenberg et al., 2010 for a nice review of the technique and its advantages; https://www.frontiersin.org/articles/10.3389/fnhum.2010.00222/full).

Response: We thank the Reviewer for recommending this highly relevant visualization and interpretation of compatibility effects. As suggested, we plotted our data (from Experiment 1 and 2 combined) in the form of a delta plots. We plotted the difference of response times between similar and dissimilar affordance conditions for compatible (1st plot) and incompatible (2nd plot) trials. What we see seems in line with what is reported by Van Den Wildenberg et al. (2010): when target and response are compatible, the mean difference between similar and dissimilar affordances changes along response times distribution. The longer the response times, the more negative the effect. This indeed suggests an increase of inhibition over time. When target and response are incompatible, we do not observe this pattern. We added the plots as a figure in the Manuscript (figure 7, page 28) as well as paragraph in the results section to introduce the delta plots, page 27, line 621: “Support for inhibition processes has been often sought in the temporal dynamics of stimulus-response compatibility effects (50). Delta plots displaying the RT difference between compatible and incompatible conditions as a function of response time distribution are typically used to this aim. The rationale is that inhibition takes time to occur and should be more reflected in the response for slower than shorter decisions, leading to changes of compatibility effects over time following a negative slope. The same visualization was applied here for distractor affordance similarity effects. As highlighted on Fig 7, delta plots of affordance similarity effects on compatible trials also show a negative slope, reflecting increased interference from similar distractors for longer response times. Such increase over time was not observed on incompatible trials. The pattern observed in the compatible condition parallels what has been reported in the literature on inhibitory control in compatibility tasks.”

We also added a small related paragraph in the discussion section (page 33, line 756): “While both the inhibition and competition hypotheses are plausible, the evolution of the effect of similarity as a function of response time distribution (Fig 7) may offer additional elements of interpretation for the type of mechanism underlying interference form distractors with similar affordances. When target and response were compatible, the difference between similar and dissimilar affordances changed along response time distribution: the effect turned more and more negative as response times increased. The temporal dynamics of the effect of affordance similarity suggest an increase of inhibition over time, as proposed by Wildenberg et al. (50). When target and response were incompatible, this temporal pattern was not observed, which further supports the inhibition hypothesis. Further investigation will be needed to better characterize the mechanisms involved in the cost entailed by similar distractor affordances and its relation to inhibition processes.”

Delta plot of the effect of Similarity for (a) compatible and (b) incompatible target and response for data from Experiment 1 and 2:

Note. mEffect corresponds to the difference of response times between the dissimilar and similar conditions as a function of response time distribution (mBin). Error bars correspond to the standard errors.

3) I’m not necessarily surprised that correct responses were not overall faster when distractors afforded an action that was incompatible, when target and response were compatible. This reminds me of Eriksen Flanker tasks (e.g., https://link.springer.com/article/10.3758/BF03203267), as one example but there are many others) – where participants are typically faster to respond when target and flakers (distractors) are associated with the same response (congruent) relative to different responses (incongruent). Perhaps there is a similar mechanism, here, whereby distractors affording responses opposite the target create competition which needs to be resolved? It might be helpful for the authors to make theoretical links to other conflict tasks, or to explain why they believe their task/mechanism to be different here.

Response: Indeed, Eriksen & Eriksen (1974) investigated the influence of distractors sharing similar or dissimilar properties with target-on-target identification. They highlighted a facilitation effect from distractors with similar visual properties on target identification. This facilitation of similar properties of distractors on target identification is specifically true when target and distractors match in terms of visual (or abstract) properties. Importantly, we found a result in the opposite direction when distractors are similar in terms of affordances with an interference effect from distractors with similar affordances on target selection. The first demonstration of this reverse pattern has been interpretated as additional evidence that the effect is action-based and not visually-based. We have added the distinction with typical flanker tasks in the introduction page 6 line 114: “Second, natural perceptual scenes are rarely composed of isolated objects but usually feature multiple objects. Without considering the evocation of affordances, the influence of distractors on target identification has been investigated for target and distractors sharing similar or dissimilar visual properties in classical flanker tasks (20). Authors usually highlight slower response times to identify the target when target and distractors shared dissimilar visual properties in comparison to similar visual properties. One may then wonder if the cost found for target identification when distractors shared dissimilar visual properties with the target may be also found when target and distractors evoke dissimilar motor properties or affordances.”

Regarding the interpretation of our effect as a result of a competition between affordances, Cisek (2007) proposed a model of action selection when multiple affordances are simultaneously evoked in an environment. We detailed this model and its predictions in the Introduction section of the manuscript (page 7, line 149) as follow: “When several affordances are simultaneously available in the environment, observers would first activate all the different possible affordances in parallel. Information would then be accumulated from various sources (e.g., sensory information about possible targets, motor information about potential reaching movements, cognitive information about goals and expected utility of actions…) in order to bias the competition and select the most relevant affordance to interact with the target object. In this framework, one may thus expect distractors with dissimilar affordances to interfere more with the processing of the target object than distractors with similar affordances. The processing of distractors with dissimilar affordances would cumulate the duration of two processes: affordance activation and affordance selection.” This model of competition between affordances predicts a cost for dissimilar affordances in comparison to similar affordances. However, we actually found the opposite result pattern with a cost for similar affordances, which is not directly compatible with the affordance competition hypothesis. Although competition could still explain the selective interference for the right hand (see our response to Reviewer 1, point 5), examination of compatibility effects across the RT distribution (see point 2 above) points more towards an interpretation in terms of inhibition processes. 

Reviewer: 3

1) line 84: I believe this matter is still very debated, so such a conclusion is not warranted

Response: We agree with the Reviewer that this interpretation is debated and compatibility effects may also be attributed to a match of abstract properties of both stimuli and response, as presented in the following paragraph of the introduction. Nonetheless, we have added some nuance to our original conclusion page 4, line 80 “Overall, the compatibility effect between the visual properties of the object and the motor properties of the response may be taken as evidence of the activation of micro-affordances from visual objects.”

The alternative interpretation of compatibility effects is also presented page 4, line 83 in the manuscript: “Yet alternative explanations of compatibility effects have been proposed, relying on the compatibility between abstract codes associated to the stimulus and the response (6–8). The stimulus could be coded as abstractly [small] or [large] in opposition to evoking precision or power grasps. Similarly, the response could also be considered as a [small] or [large] response independently of the type of grasp. A compatibility effect would arise when the abstract codes of both the stimulus and response match. However, recent findings indicated that the combination of specific types of task (e.g., a task relevant for action) and response (e.g., reach-and-grasp response) could favour the activation of action components (9). The affordance hypothesis is also legitimated by neurophysiological studies highlighting an activity of the motor system during the perception of manipulable objects, independently of the SRC paradigm (10–13). In consequence, although abstract coding may be frequently at play in compatibility effects, it does not completely rule out the existence of affordance activation in some specific situations.”

2) line 249: they were informed about the aim of the study: what kind of details were provided?

Response: Participants were informed that the experiment focused on visual perception of objects and aimed to understand how we categorize objects. At this stage, the task was also quickly presented. We informed participants that they would see scenes of two objects, and they would have to determine if objects are kitchen utensils or tools. More details about the task were given once the participants launched the experiment. 

We added some details about these aspects in the manuscript, page 11, line 244: “Participants were informed about the study by receiving an automatic email from Prolific with the experiment details if they met the experiment inclusion criteria. They were aware that the study focused on visual perception of objects and aimed to understand how we categorize object among distractors. When clicking on the link to the study, they were again informed about the objective of the study. They were also informed about the task, namely that they will see scenes of two objects and will have to determine if objects are kitchen utensils or tools.”

3) line 277: total stimuli 96 stimuli / 8 configurations = 12 stimuli per configuration. This might partially be the reason the lack of effects in the the two (separate) experiments….is there a reason behind this choice? 12 stimuli per “design cell” are very few.

Response: We agree with the Reviewer that 12 stimuli per configuration is few and it participates to the statistical power of the experiment. However, it is typical with meaningful and well controlled nonconflictual handled stimuli (Fairchild et al., 2021; Masson-Carro et al., 2016). Stimulus repetition may reduce differences between conditions, as the task becomes easier (ceiling effect). Moreover, we wanted to keep the duration of the task short to prevent (as much as possible) an influence of fatigue or decrease of attention on RTs. Thus, the number of stimuli was chosen to best compromise between statistical power and data variability. 

References:

Fairchild, G. T., Marini, F., & Snow, J. C. (2021). Graspability Modulates the Stronger Neural Signature of Motor Preparation for Real Objects vs. Pictures. Journal of Cognitive Neuroscience, 33(12), 2477-2493.

Masson-Carro, I., Goudbeek, M., & Krahmer, E. (2016). Can you handle this? The impact of object affordances on how co-speech gestures are produced. Language, cognition and neuroscience, 31(3), 430-440.

4) Why are the trials so short? the transition between the 3 scenarios is extreme.

Response: While the duration of the empty scene and fixation cross are pretty short (500ms each), the duration of the stimulus presentation was not particularly short. Indeed, participants had as much time as needed to respond to the task, with the stimulus displayed until the participants answered. In addition, the task was very simple (key presses on the side of the target) with participants having their hands ready on the keyboard. With such a simple and short responses, not much delay was needed between trials. The reaction times of participants were also quite short (mostly between 800ms and 900ms; see Appendix 3). We also pretested the experiment in the laboratory on naïve participants before putting it online in order to find the best time course. Again, we chose the best compromise to ensure that the task was paced enough so that participants would not be distracted between trials but could still perform the task correctly. 

We added some clarification about the time course of the trials page 14 line 327: “They had to answer as accurately and quickly as possible on the keyboard by pressing simultaneously the “e” and “c” keys if the target object was on the left or the “i” and “n” keys if object was on the right. Participants had as much time as needed to respond to the task.”

We also added a foot note page 15 regarding the laboratory pre-test of the experiment before conducted it online: “Timing of the trial procedure was pre-tested in the laboratory before conducting the online experiments.”

5) exclusion of responses >4000 ms…It looks like even 2500 ms is enough to provide a response…Why 4000ms?

Response: This first global trimming at 4000ms is only undergone to remove any completely aberrant response times (for example due to participants coughing or sneezing during a trial or loosing focus for several seconds) that would affect computation of individual standard deviations for the main trimming procedure. We subsequently performed a trimming of response times per subject and condition at 2.5 standard deviation from the mean response times that allowed us to better remove outlier trials. 

We added these details in the manuscript, page 16, line 372: “A global trim was undergone by excluding RT inferior to 200 ms and RT superior to 4000 ms, to remove any aberrant responses. We then excluded RTs superior to 2.5 standard deviations from the mean RT of each participant in each condition (affordance similarity x target and response compatibility x task version).”

6) The power analysis reported indicates that 120 participants were necessary to get an effect. However, participants resulting from exp 1 and 2 are ~60. 

Response: Regarding the number of participants recruited, we recruited 146 participants for Experiment 1 (See Page 11, in the participants section: “One hundred and forty-six participants (57 women) between 18 and 40 years old were recruited”) and 137 for Experiment 2 (See Page 21, in the participants section: “One hundred and thirty-seven participants (68 women) between 18 and 40 years old were recruited”). 

Moreover, the authors mentioned the power but not the alpha level (<0.05?) and the tails of the hypothesis (which I assume is 2).

Response: We forgot to specify this important information in the manuscript. Indeed, we have a two-tail hypothesis. Regarding the different indexes of the power analysis, we considered a beta level of 0.2 and therefore a power (1-beta) of 0.8 for an alpha level of 0.05. 

We specified this information on page 12, line 263: 

“To guarantee a sufficient statistical power for a two-tails hypothesis (β = 0.2; power (1- β) = 0.80; α = 0.05), about 120 participants were anticipated.”

7) Instead of changing the data with the 2SD rule (which is not optimal for outlier detection), the authors might use the medians instead of the means so that big outliers should not affect the central tendency measure. Considering the small number of trials per cell I think this is the more elegant solution. Otherwise check Jones 2019 https://link.springer.com/article/10.3758/s13414-019-01726-3 for other strategies.

Response: We thank the reviewer for this comment. We actually initially thought about using the Median Absolute Deviation (MAD) for RTs trimming but did not finally choose this method because of the large proportion of data it would remove. Please find below the percentage of trials removed after trimming and the response times distribution after trimming and removing outlier participants for Experiment 1 and 2:

Experiment 1: Experiment 2: 

8 % of trials removed. 9% of trials removed. 

RTs distribution with MAD method RTs distribution with MAD method:

Although we might gain a little of skewness reduction with MAD compared to the mean/SD trimming method (see also our response to point 1 raised by Reviewer 2), the percentage of trials removed is close to 10% (compared to 3% with the current method). Thus, we decided to keep the mean/SD trimming. 

Note however that mixed model results are very consistent between the two trimming methods. This has been added in a footnote on page 17. You may refer to the comment 1 of the Reviewer 2 for a more detailed comparison between RTs distribution for different trimming methods and data transformations.

8) The authors found an effect, that partially fit their hypothesis. This effect however arises from the total sample of participants who participated in both experiments. I would like the authors to calculate and report the effect size of their results.

Response: We acknowledge that we did not initially report effect sizes of our results. The reason behind this choice is that effect sizes for linear mixed model effects are not straightforward to compute and interpret. This is because effect sizes measure in mixed-effect models capture the magnitude of fixed effects while taking into account random effects random effects (random effects contribute to the total amount of variance explained). Brysbaert and Stevens (2018) presents a review of effect sizes in different study designs as well as an alternative to Cohen d for mixed-effect models, initially proposed by Westfall et al. 2014. 

Another alternative would be the one proposed by Westfall et al., 2014: The Westfall d is computed as the mean difference divided by the standard deviation of the sum of the variance of random parameters (here participants and items). It is now implemented in the eff_size function from emmeans R package, which is adapted for mixed-effects models. In the revised version, we reported Westfall d for the different contrasts presented. We also present this effect size measure on page 18 line 401: “Effect sizes were computed as Wesftall’s d, an alternative to Cohen’s d suitable for linear mixed model effects (46,47). Westfall’s d measures were computed with the eff_size function of the emmeans package (48).”

Note however that effect sizes found with this method are usually a lot smaller than typical effect sizes and should not be interpreted as Cohen d. 

References:

Brysbaert, M., & Stevens, M. (2018). Power analysis and effect size in mixed effects models: A tutorial. Journal of cognition, 1(1).

Westfall, J., Kenny, D. A., & Judd, C. M. (2014). Statistical power and optimal design in experiments in which samples of participants respond to samples of stimuli. Journal of Experimental Psychology: General, 143(5), 2020.

9) Curiously the authors did not find an affordance effect independently from the distractors (i.e. produced by the handle orientation itself, the compatibility). Did the authors expected that?

Response: Based on previous results studying the evocation of single affordances using compatibility paradigms, we expected a main effect of compatibility between target and response (Bub et al., 2015; Tucker and Ellis 2001). However, there is not sufficient literature to know exactly how strongly the distractor affordances would impact the perception of the affordance of the target. Therefore, even though we hypothesized that a compatibility effect would arise, it is not surprising that it did not appear irrespective of the affordances of the distractors and the response hand, but rather modulated by these factors. In accordance with the Reviewer’s comment, a discussion of the absence of compatibility effect has been added in a limitation paragraph on page 33, line 767: “This study provides novel evidence on the influence of the similarity of affordances evoked by multiple objects on object perception. Yet, it is important to emphasize some limitations to our results. First, we neither observe an overall effect of target-response compatibility nor an influence of distractor affordances on object selection independently of response hand. The impact of target-response compatibility on response times may have not been strong enough to overcome the modulations entailed by the other factors of interest, namely response hand and distractor affordances. Second, the general advantage for the left hand/hemifield remains difficult to explain, although the effect probably originates at the motor rather than visuo-attentional level. Finally, while the interaction between target-response compatibility, distractor affordance similarity and response hand was significant, the effect size for this interaction was small. Further investigations would be helpful to obtain a clearer view of the pattern of affordance effects reported in this present study, especially how the effect of multiple affordances on object selection is modulated by the response hand. Follow up studies could benefit from a greater number of trials per condition to maximize statistical power.”

References:

Bub, D. N., Masson, M. E., & Lin, T. (2015). Components of action representations evoked when identifying manipulable objects. Frontiers in Human Neuroscience, 9, 42.

Tucker, M., & Ellis, R. (2001). The potentiation of grasp types during visual object categorization. Visual cognition, 8(6), 769-800.

10) The results part is not written properly in my opinion. It would be much better if the main effects are described before the interactions, and the interactions with a lower number of factor would be described before the ones with a higher number of factors (two-way before three way).

Response: Reporting main effects before interaction could lead to errors in the interpretation of the effects (Nieuwenhui et al. 2011; Sawilowsky et al., 2007). So, we prefer to present the highest effect of interest first. 

References:

Nieuwenhuis, S., Forstmann, B. U., & Wagenmakers, E. J. (2011). Erroneous analyses of interactions in neuroscience: a problem of significance. Nature neuroscience, 14(9), 1105-1107.

Sawilowsky, S., & Sawilowsky, S. S. (2007). Effect sizes, simulating interaction versus main effects, and a modified ANOVA table. Real Data Analysis, 191.

11) It would also be advisable to properly and explicitly describe the design (es 2x2x2 with three within independent variables, including factor1(level a, level b), factor2 (level a level b) etc).

Response: We clarified the design in the revised version of the manuscript, in the 1st paragraph of page 17, line 388: “the fixed effect factors included i) the Similarity of handle affordances between target and distractor (similar and non-similar), ii) the Compatibility between target orientation and response (compatible, incompatible) and iii) the Response Hand corresponding to the target location (left, right).”

12) I think that in general the manuscript could be streamlined: es.: line 318 some of the description has already been mentioned before

Response: We thank the Reviewer for this suggestion, we removed the redundancies and streamlined some parts of the manuscript, namely the section about:

-competition between multiple affordances evoked by a single object (page 5, line 101).

-empirical data on the evocation of multiple affordances evoked by different objects (page 8, line 175).

13) In the discussion little or no space has been given to the unexpected result of related to the faster left hand responses compared to right hand ones.

Response: We agree that the manuscript was lacking this very important discussion (which was also highlighted by Reviewer 1, comment 7). 

We added a paragraph in the Discussion section of the manuscript (page 29, line 675) about this effect: “In the first experiment, we failed to observe an effect of distractor affordances on target processing. However, a strong general advantage of the left non-dominant response hand also corresponding to the left location of the target was found. In the second experiment, we aimed at determining whether this general left advantage could be due to a visuo-attentional bias. By introducing catch-trials requiring attention to both objects of the scenes, we expected a reduction of this putative bias. Catch-trials did not attenuate the left advantage, suggesting that the effect originated in the response selection rather than in a visuo-attentional bias. The reasons underlying overall faster motor responses with the left, non-dominant hand regardless of handle affordances of target and distractor objects remain to be elucidated. The most parsimonious interpretation relies on the biomechanical constraints of the hand postures involved in the key press responses, that might be not completely equivalent between left and right hands (E-C vs I-N presses respectively).”

We also further discussed response hand on page 31, line 724: “Furthermore, the fact that the impact of distractor affordances on object selection was restricted to responses made with the dominant right hand is particularly interesting. Numerous studies have highlighted differences between the dominant right hand and the left hand in action selection tasks, when objects evoke affordances, with the dominant right-hand generally more sensitive to affordances (49,51,52). This greater susceptibility of the dominant hand to affordance effects might be related to the better performance of the dominant right hand in comparison to the left hand in motor coordination, motor execution and motor planning (51,53). Yet in the present study, the right dominant hand was more sensitive to affordances effects despite overall faster motor responses with the left hand, suggesting that the potentiation of grasp components from visual objects may be relatively independent from general response speed.”

14) line 592: there is an asterisk which should not be there I believe :-)

Response: Indeed, the asterisk should not be there. We removed it. 

15) line 614: response selection (instead of response hand)

Response: We have changed this sentence accordingly page 29, line 680: “Catch-trials did not attenuate the left advantage, suggesting that the effect originated in the response selection rather than in a visuo-attentional bias.”

16) line 618: “identify” might be misleading: one might intepret it as the motor representations facilitated by the affordance effect are necessary to identify the object as a kitchen tool or an utensil. This is not necessarily the case. I would use “select” instead (also in other sections of the manuscript).

Response: We agree and have changed the sentence accordingly page 30 line 688: “When participants had to respond with their right dominant hand and when target and response were compatible, they were slower to select the target object presented with a distractor object with similar compared to dissimilar affordances.”

We have also applied this change throughout the manuscript. 

17) line 297 and 628: one can turn the argument the other way around: this strategy might have been detrimental in letting the effects emerge: the shape of the responding fingers might be quite important, but the one depicted in the figure is quite far from the final position that could be adopted by a person grasping one of these objects. It resambles more a precision grip with two fingers. The visual resamblance between the participants’ fingers shape and the one that would have used in order to grasp the object does not necessarily help the facilitation of the motor representation of a whole hand grasp, on the contrary, it might have inhibited the “object-grasp hand shape”, because a pinch-like gasping fits different type of objects (seldom one would grasp a whisker with hands shaped like those in figure 2). In general, when looking for automatic visuomotor effects, the best motor responses to be selected for the task are those that fit/resamble the most the object/action observed (a general argument for stimulus response compatibility effects: Kronblum 1990 10.1037/0033-295X.97.2.253; Barchiesi & Cattaneo 2015 10.1016/j.neuropsychologia.2015.01.030 on automatic visuomotor effects).

Response: We agree with the reviewer’s comment. While our initial intention has been to mimic the grasping behavior, the visual resemblance between the hand posture required to grasp the object and the shape of the responding fingers is not optimal. However, the grasping position required to respond is not typical and required deeper control than simple button press. Besides, in the present version of the manuscript, we focused our purpose on orientation affordances and how handle direction evoked a left/right response. While a potential mismatch between the hand posture required to grasp the object and the shape of the responding fingers could have diminished the evocation of size affordances, it cannot explain the result pattern regarding the similarity and compatibility for orientation affordance. We added details on the shape of the hand related to grasp aperture page 13, line 297: “Even though the grasp aperture of the response was not necessarily tuned to the object, the size of the grasp was not critical to our design as we manipulated the affordance compatibility between handle left/right orientation and left/right response and not between the size of the object and the size of the grasp response.”

18) line 641: The task lacks a control for the affordance effect. For example, if the effect is genuine (and not spatially driven by the handle directions for example), then, the authors should have expected a modulation of the effects if the shape of the fingers was changed, or if abstract oriented stimuli were added as control condition. The authors claim that the “interference by similarity effect” found combining the two experiments rules out the “abstract spatial effects” explanation even though the objects are oriented shapes. According to the authors this claim is strengthened by the result obtained, i.e., that the interference effect is found only for the right hand. Again the argument in favour of the results can be turned against: the authors did not predict a right hand effect specifically for the compatibility x distractors effect. The interpretation of the three-way interaction is a post-hoc one. So, who knows whether using other abstract stimuli as a control condition the authors might have got similar effects as those obtained with the objects…

Response: We acknowledge that we did not test the abstract coding hypothesis directly. The reason why we are quite confident that our results reflect the evocation of affordances and not an abstract coding is because of the direction of the effect found. The direction of the expected and observed effect of affordance similarity is in the opposite direction as what the abstract coding would predict. Previous studies tested the influence of visual properties of distractors on the identification of a target object with similar or dissimilar visual properties (Eriksen & Eriksen., 1974). They highlighted a facilitation effect of distractor with similar visual properties as the target-on-target identification. This effect is also reported in the study of Pavese and Buxbaum (2002). In the present studies, if the compatibility effects arose from a match of abstract or visual properties only, we would have expected an effect in the direction of a facilitation of similar affordances on target selection. However, we found the opposite effect with an interference of similar affordances, which is why we can reasonably interpret the effect as reflecting the evocation of affordances.

These aspects are discussed on page 31, line 719 of the manuscript. Note that we did not rule out the abstract coding hypothesis, which would require direct testing. We rather concluded that the pattern of results “would be difficult to explain from an abstract coding perspective” 

We have added the distinction with studies testing the influence of visual properties of distractors on the identification of a target object with similar or dissimilar visual properties (Eriksen & Eriksen., 1974) in the introduction page 6, line 114: “Second, natural perceptual scenes are rarely composed of isolated objects but usually feature multiple objects. Without considering the evocation of affordances, the influence of distractors on target identification has been investigated for target and distractors sharing similar or dissimilar visual properties in classical flanker tasks (20). Authors usually highlight slower response times to identify the target when target and distractors shared dissimilar visual properties in comparison to similar visual properties. One may then wonder if the cost found for target identification when distractors shared dissimilar visual properties with the target may be also found when target and distractors evoke dissimilar motor properties or affordances.”

Eriksen, B. A., & Eriksen, C. W. (1974). Effects of noise letters upon the identification of a target letter in a nonsearch task. Perception & psychophysics, 16(1), 143-149.

Pavese, A., & Buxbaum, L. J. (2002). Action matters: The role of action plans and object affordances in selection for action. Visual cognition, 9(4-5), 559-590.

19) line 650: ok, but if this is true, then how do the authors explain the left hand speed? I would expect that right hand ideal conditions (dissimilar distractor compatible handle) would produce the fastest reaction times within the experiment…which is not the case…unless

Response: Indeed, the speed of left-hand responses in comparison to right-hand ones is surprising. However, the abstract coding hypothesis does not appear completely plausible to explain the left-hand speed. If compatibility effects reflected a coding of abstract properties, we would expect compatibility effects to appear also for left-hand responses, which is not the case here. Further investigation of response hands effects is presented in response to the 5th comment of Reviewer 1, where we conducted supplementary analyses on compatibility effects as a function of affordance similarity and response hand. 

20) line 658: it is not necessarily an inhibition mechanism…it could , be a competition mechanism right?

Response: We agree that some type of competition mechanism may been at play. You may refer to the comment 5 of the Reviewer 1 for a detailed discussion about the potential competition between similar affordances. We added a paragraph in the Discussion section page 32, line 744 to discuss this specific competition interpretation of our effect: “An alternative interpretation based on competition may explain the interference effect from distractors with similar affordances during target selection reported here. When analyzing data from the perspective of compatibility effects, we observed greater compatibility effects when target and distractor evoked dissimilar affordances in comparison to similar affordances for right hand responses. One could argue that when affordances of target and distractor are dissimilar, only the affordance of the target object is compatible with the response. In that case scenario, no competition occurs between the affordances of the target and distractor and a classical compatibility effect is observed. In contrast when target and distractor evoke similar affordances, both handle affordances compete for right hand selection, which leads to slower compatible responses and therefore a reduction of compatibility effects. Such competition would not be visible for left hand responses, as the left hand would be less sensitive to compatibility effects overall (49,51,52).”

21) When in time do the authors expect an affordance effect? the average responses were very delayed (~800 ms). I was wondering whether the task was so difficult that the participants responses were not very influenced anymore by the affordances. On other visuomotor effects these are very early in time (Barchiesi & Cattaneo 2015 10.1016/j.neuropsychologia.2015.01.030).

Response: The response times may be quite long considering button press responses and the timing of visuomotor effects. We do not think that the task was particularly difficult, considering that the accuracy was not at ceiling but still quite high (M = 0.87 and M = 0.90 respectively for experiments 1 and 2). In addition, if we look at the effects in the study by Pavese and Buxbaum (2002), the mean response times (for button press and grasp responses) were between 800ms and 1000ms, which is in adequation with the response times we found. These pretty late effects may be due to the processing of two objects evoking two affordances in comparison to previous studies where only a single object (and single affordance) was perceived. This difference may also be due to the selection process toward the target object that needs to be undergone when two objects are perceived, which again is not the case for single object perception. Also, in our experiment, the specific hand posture required for the button presses may have slowed down responses in comparison to typical button press responses. 

Pavese, A., & Buxbaum, L. J. (2002). Action matters: The role of action plans and object affordances in selection for action. Visual cognition, 9(4-5), 559-590.

22) I would like the authors to add a limitation paragraph including:

the partial confirmation of their hypothesis (no compatibility main effect, no interaction between compatibility and distractors) left hand responses are faster in general (and add speculations about it) the (most likely) small effect size of their results in terms of distractor similarity how the small number of trials per cell could have been accounted for the lack of fully expected effects

Response: In accordance with reviewer comment, a limitation paragraph has been added page 33, line 767: “This study provides novel evidence on the influence of the similarity of affordances evoked by multiple objects on object perception. Yet, it is important to emphasize some limitations to our results. First, we neither observe an overall effect of target-response compatibility nor an influence of distractor affordances on object selection independently of response hand. The impact of target-response compatibility on response times may have not been strong enough to overcome the modulations entailed by the other factors of interest, namely response hand and distractor affordances. Second, the general advantage for the left hand/hemifield remains difficult to explain, although the effect probably originates at the motor rather than visuo-attentional level. Finally, while the interaction between target-response compatibility, distractor affordance similarity and response hand was significant, the effect size for this interaction was small. Further investigations would be helpful to obtain a clearer view of the pattern of affordance effects reported in this present study, especially how the effect of multiple affordances on object selection is modulated by the response hand. Follow up studies could benefit from a greater number of trials per condition to maximize statistical power.”

---

## [Decision Letter · Decision Letter 1]

7 Aug 2023

Too much to handle? Interference from distractors with similar affordances on target selection for handled objects

PONE-D-23-04726R1

Dear Dr. Kalénine,

We’re pleased to inform you that your manuscript has been judged scientifically suitable for publication and will be formally accepted for publication once it meets all outstanding technical requirements.

Kind regards,

Luigi Cattaneo, MD, PhD

Academic Editor

PLOS ONE

Additional Editor Comments (optional):

Reviewers' comments:

Reviewer's Responses to Questions

**Comments to the Author**

1. If the authors have adequately addressed your comments raised in a previous round of review and you feel that this manuscript is now acceptable for publication, you may indicate that here to bypass the “Comments to the Author” section, enter your conflict of interest statement in the “Confidential to Editor” section, and submit your "Accept" recommendation.

Reviewer #1: (No Response)

2. Is the manuscript technically sound, and do the data support the conclusions?

Reviewer #1: Yes

3. Has the statistical analysis been performed appropriately and rigorously? 

Reviewer #1: Yes

4. Have the authors made all data underlying the findings in their manuscript fully available?

Reviewer #1: Yes

5. Is the manuscript presented in an intelligible fashion and written in standard English?

Reviewer #1: Yes

6. Review Comments to the Author

Reviewer #1: I think the authors have adequately addressed and discussed most of my points and questions and I thank them for their careful point-by-point responses. Therefore, I have no further comments.

7. PLOS authors have the option to publish the peer review history of their article (what does this mean?). If published, this will include your full peer review and any attached files.

Reviewer #1: No

---

## [Editor Report · Acceptance letter]

21 Aug 2023

PONE-D-23-04726R1 

Too much to handle? Interference from distractors with similar affordances on target selection for handled objects. 

Dear Dr. Kalénine:

I'm pleased to inform you that your manuscript has been deemed suitable for publication in PLOS ONE. Congratulations! Your manuscript is now with our production department. 

Kind regards, 

on behalf of

Dr. Luigi Cattaneo 

Academic Editor

PLOS ONE